# Directed evolution of and structural insights into antibody-mediated disruption of a stable receptor-ligand complex

Luke F. Pennington [1,2,3], Pascal Gasser [4,5], Silke Kleinboelting[1], Chensong Zhang[6], Georgios Skiniotis [1,6], Alexander Eggel [4,5] & Theodore S. Jardetzky [1,2,3 ✉]

Antibody drugs exert therapeutic effects via a range of mechanisms, including competitive inhibition, allosteric modulation, and immune effector mechanisms. Facilitated dissociation is an additional mechanism where antibody-mediated "disruption" of stable high-affinity macromolecular complexes can potentially enhance therapeutic efficacy. However, this mechanism is not well understood or utilized therapeutically. Here, we investigate and engineer the weak disruptive activity of an existing therapeutic antibody, omalizumab, which targets IgE antibodies to block the allergic response. We develop a yeast display approach to select for and engineer antibody disruptive efficiency and generate potent omalizumab variants that dissociate receptor-bound IgE. We determine a low resolution cryo-EM structure of a transient disruption intermediate containing the IgE-Fc, its partially dissociated receptor and an antibody inhibitor. Our results provide a conceptual framework for engineering disruptive inhibitors for other targets, insights into the failure in clinical trials of the previous high affinity omalizumab HAE variant and anti-IgE antibodies that safely and rapidly disarm allergic effector cells.

[1] Department of Structural Biology, Stanford University School of Medicine, Stanford, CA 94305, USA. [2] Progam in Immunology, Stanford University School of Medicine, Stanford, CA 94305, USA. [3] Sean N. Parker Center for Allergy Research at Stanford University, Stanford, CA 94305, USA. [4] Department of Rheumatology and Immunology, University Hospital Bern, Bern, Switzerland. [5] Department of BioMedical Research, University of Bern, Bern, Switzerland. [6] Department of Molecular and Cellular Physiology, Stanford University School of Medicine, Stanford, CA 94305, USA. ✉email: tjardetz@stanford.edu

Therapeutic antibodies represent a dominant and growing percentage of new drugs, and new technologies for antibody discovery and engineering have advanced rapidly. Antibodies can exert their therapeutic impact through a variety of mechanisms. Some of these mechanisms rely solely on the binding, inhibition or blocking of their target, while other mechanisms rely on accessory functions mediated by antibody Fc-domains, such as the activation of complement, antibody-dependent cellular cytotoxicity or phagocytosis[1,2].

In addition to these well-appreciated mechanisms of action, some antibodies and antibody alternatives have been shown to act through kinetic mechanisms that involve the facilitated dissociation of targeted complexes[3–6]. During facilitated dissociation "disruptive" inhibitors engage targets that are sequestered in stable macromolecular complexes, accelerate their dissociation, and then inactivate them, rather than simply blocking interactions between proteins (Fig. 1a, b). Structural studies of disruptive inhibitors and pairs of antibodies that can displace one another suggest that disruption is mediated by small regions of predicted steric conflict, in contrast to strictly competitive inhibitors with large regions of steric conflict (Fig. 1c–e). Conformational rearrangement of protein complexes and/or changes in the disruptive inhibitor binding pose are likely necessary to resolve these small conflicts, allowing for the formation of an unstable trimolecular complex which rapidly dissociates (Fig. 1b). Unlike allosteric inhibitors that may also destabilize complexes, disruptive agents sterically compete with receptor, do not require long-range conformational changes within a protein domain, and may therefore be applicable to many protein-protein interactions.

We have previously studied the disruptive activities of antibodies and Designed Ankyrin Repeat Proteins (DARPins) directed against IgE antibodies[3,4,7,8]. IgE antibodies underly allergic diseases such as allergic asthma, chronic urticaria (hives), chronic sinusitis and food allergy[9–11]. Most antibody isotypes bind their Fc-receptors weakly and depend on the formation of avid immune complexes for stable interactions, however IgE binds its high-affinity receptor (FcεRI) with a $K_D$ of ~100 pM. Mast cells and basophils are therefore coated with IgE antibodies and primed for activation prior to antigen exposure and this priming persists for months[12]. Therapeutic anti-IgE antibodies (e.g., omalizumab and ligelizumab) were specifically selected to compete with IgE:FcεRI interactions, and not target IgE receptor complexes. Thus their rate of action is limited by the extremely long half-life of IgE:FcεRI complexes[8]. Omalizumab and ligelizumab take weeks to achieve maximal response in allergic patients[13]. Although this limitation has not stopped the application of anti-IgE in the treatment of chronic allergic diseases or gradual allergen desensitization, the therapeutic effect of these agents could be significantly accelerated if they targeted and inactivated existing IgE:FcεRI complexes.

Omalizumab displays weak disruptive activity at high concentrations, while ligelizumab does not display any disruptive activity[3,4], despite the ~100-fold higher affinity of ligelizumab for IgE. In contrast a class of DARPin based anti-IgE agents, including the DARPin E2_79, are more effective disruptors of IgE:FcεRI complexes while having similar affinities to omalizumab[4]. To classify these disruptive agents, we developed the concept of disruptive "efficiency" ($ID_{50}/K_D$), which relates the half-maximal disruptive concentration ($ID_{50}$) to the affinity for free IgE ($K_D$)[4]. This disruptive efficiency is effectively a representation of the free energy barrier imposed on the anti-IgE inhibitor for binding to the receptor complex by the steric overlap with the receptor, and improvements in efficiency relate to a decrease in this free energy barrier (Supplementary Fig. 1a). This framework, together with structural studies[3,7,8], has provided insight into this mechanism of inhibition and shown that the degree of steric overlap relates to the efficiency of disruption. Nevertheless, our understanding of this process has been limited by the transient nature of the intermediates that must form between the inhibitors and complexes prior to disruption. Furthermore, no prior studies have developed a systematic experimental pathway to select or evolve more efficient disruptive agents or to understand parameters beyond steric overlap that modulate disruption. Although our studies with DARPins have previously demonstrated that anchoring disruptive anti-IgE domains to non-inhibitory anti-IgE binders can dramatically enhance potency[4,14], these efforts did not engineer disruptive domains themselves or attempt to systematically study the mechanism of disruption.

We sought to provide an experimental and conceptual foundation for selecting and engineering antibodies with potent disruptive activities by reengineering the weak disruptive activity of omalizumab. We developed a yeast display "efficiency" screen to evolve omalizumab's disruptive activity and explored biophysical and structural features associated with its ability to release IgE from its receptor. To gain direct insight into the structural intermediates formed during facilitated dissociation, we solved the cryo-EM structure of an omalizumab variant bound to a partially dissociated IgE:FcεRI complex, using disulfide-stabilized IgE-Fc:FcεRI complex. This structure represents a "transition state" intermediate along the disruption pathway and reveals unanticipated interactions outside of the known antibody:antigen interface that modulate both affinity and the rate of disruption. We observe that the dwell time of disruptive inhibitors on intact complexes prior to disruption relates to their efficiency and is a critical parameter in the safe removal of IgE from human allergic effector cells. Finally, we show that at therapeutically relevant doses high-affinity disruptive omalizumab variants completely desensitize basophils from allergic donors in hours, where omalizumab has no inhibitory effect. These studies show how kinetically active disruptive antibodies can be used to accelerate therapeutic efficacy, provide a structural and conceptual foundation for understanding disruptive antibody inhibition, and establish a pathway to engineering disruptive antibodies for other diseases.

## Results

**Omalizumab disruptive potency is modulated by affinity and conformational flexibility.** To facilitate omalizumab engineering, we initially assessed the disruptive activities of soluble omalizumab Fab and single chain variable fragment (scFv) proteins (Supplementary Fig. 1b–c). We used a biochemical disruption assay using biotinylated-IgE-Fc$_{2–4}$ (bIgE-Fc$_{2–4}$) and FcεRIα-conjugated polystyrene beads to evaluate disruptive potency ($ID_{50}$) (Fig. 1f). In this assay no spontaneous dissociation is observed at low anti-IgE concentrations (Fig. 1f) or with off-target antibody controls (Supplementary Fig. 1d). Incubation with a non-competitive anti-IgE, DARPin E3_53, weakly promotes dissociation of IgE at high concentrations as previously described (Supplementary Fig. 1d)[14]. The omalizumab scFv exhibited higher affinity, secondary to a slower dissociation rate ($k_d$), and a twofold improvement in $ID_{50}$ for complex disruption as compared to the Fab (Fig. 1g and Supplementary Fig. 1e, Supplementary Tables 1 and 2). However, in these studies disruption efficiency ($ID_{50}/K_D$) remained unchanged as compared to the more efficient disruptive anti-IgE DARPin E2_79 (Fig. 1k, Supplementary Table 2). These results are consistent with prior studies on the omalizumab scFv and omalizumab Fab variants that showed small improvements in affinity lead to moderate improvements in disruption, although the $ID_{50}$ of these variants was not calculated to compare efficiencies[15,16]. To graphically

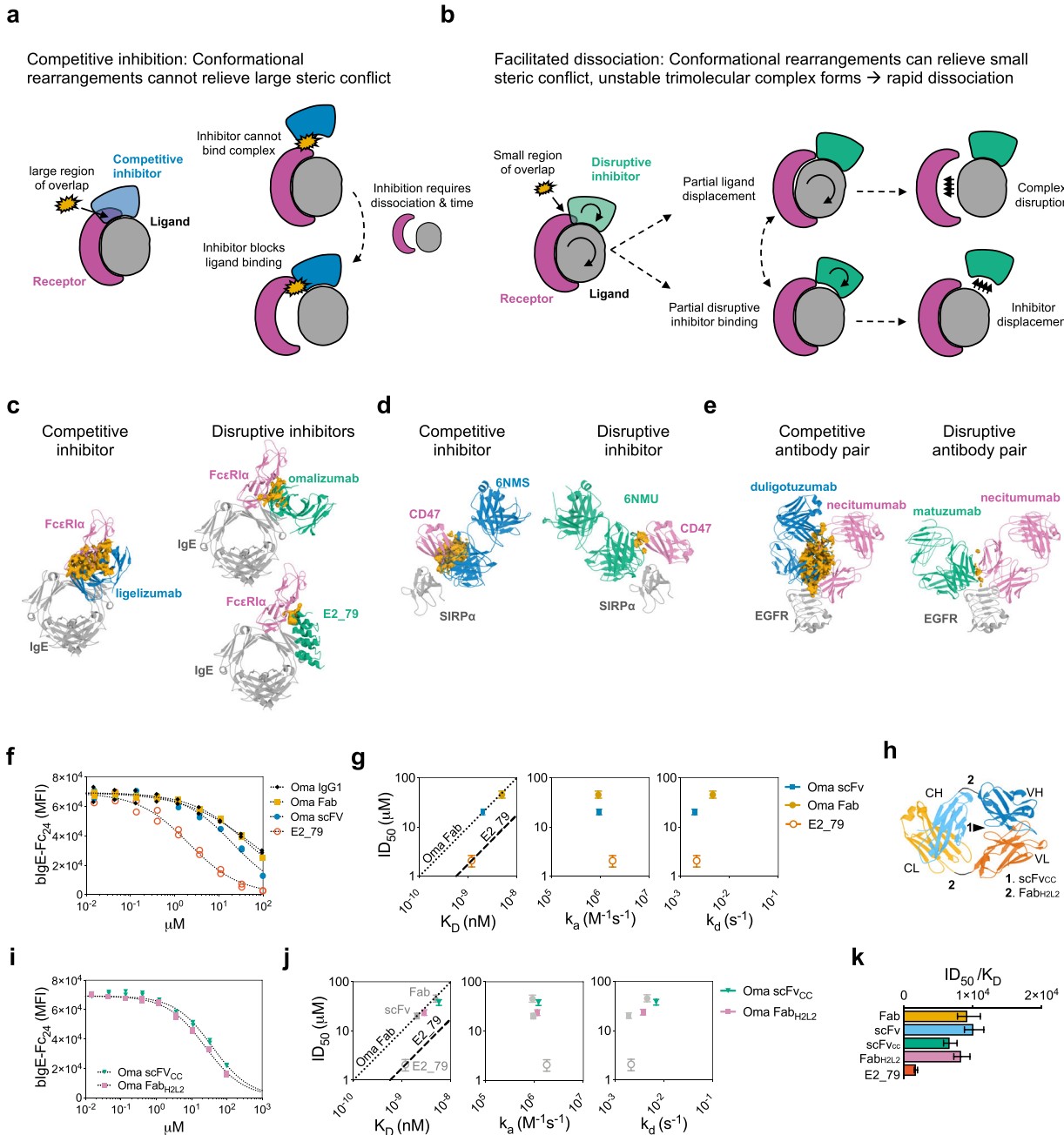

**Fig. 1 Omalizumab disruptive potency is modulated by affinity and conformational flexibility. a** Schematic of competitive inhibition. **b** Schematic of facilitated dissociation. Comparison of the predicted steric overlap volume (orange) from structural alignments of known inhibitors or disruptors of IgE:FcεRIα complexes (**c**), SIRPα:CD47 complexes (**d**), or EGFR:necitumumab antibody complexes (**e**). The orange volume represents the overlapping regions of maps generated for the indicated molecules (e.g., omalizumab and FcεRIα) in superimposed models (e.g., omalizumab:IgE and IgE:FcεRIα). Competitive inhibitors are depicted in blue and disruptive inhibitors in green. The binding target of the inhibitors are gray and the receptor or displaced molecules are magenta. **f** Bead-based disruption assay with indicated anti-IgE agents and fitted dose response curve (dotted line). **g** $ID_{50}$ with 95% CI from curve fits in **f** vs. $K_D$, $k_a$, or $k_d$ by variant. The efficiency ratio for omalizumab Fab and E2_79 is plotted as benchmarks for other variants (dotted and dashed line). **h** Schematic of conformational mutations in omalizumab Fab and scFv. **i** Bead-based disruption assay with omalizumab conformation variants and fitted dose response curve (dotted line). **j** $ID_{50}$ with 95% CI from fits in **i** vs. $K_D$, $k_a$, or $k_d$ by variant. **k** Histogram of efficiency ratio ($ID_{50}/K_D$) for each variant with error bars reflecting 95% CI from fits in **f** and **i**. Source data are provided as a Source Data file.

represent the relationships between affinity ($K_D$), potency ($ID_{50}$), and efficiency, we plotted $ID_{50}$ vs. $K_D$ with lines whose slopes ($ID_{50}/K_D$) correspond to the efficiencies of the original omalizumab Fab and DARPin E2_79 as benchmarks (Fig. 1g)[3]. Functional differences between Fabs and scFvs are common, as each format can impose restrictions on the relative orientation of variable heavy (VH) and variable light (VL) domains. To test this

possibility, we expressed two additional constructs, a constrained scFv with a disulfide bond at the VH:VL interface (scFv$_{CC}$) and a Fab construct with two glycine insertions at the elbow of the Fab heavy and light chains (Fab$_{H2L2}$)[17] (Fig. 1h). Consistent with our hypothesis the scFv$_{CC}$ mutant exhibited a $K_D$ and $ID_{50}$ similar to the Fab, whereas the Fab$_{H2L2}$ construct showed improvements in $ID_{50}$ and $K_D$ consistent with the scFv (Fig. 1i and Supplementary

Fig. 1e, Supplementary Tables 1 and 2). However, the efficiency of disruption ($ID_{50}/K_D$) remained similar (Fig. 1j and k, Supplementary Table 2). Together these data support the hypothesis that disruptive potency can be modulated by changes in binding kinetics and intrinsic structural features (e.g., conformational flexibility), while larger changes to disruptive efficiency require more substantial changes in binding behavior or molecular structure (e.g., E2_79 vs omalizumab).

**Directed evolution of omalizumab variants using a disruptive efficiency screen.** The omalizumab scFv and H2L2 variants suggest that antibody affinity and disruptive potency could increase proportionally and track linearly along the same efficiency line (Fig. 2a). However, we hypothesized that directed evolution of disruptive inhibitors could improve or impair affinity and disruptive potency independently. For example improving affinity might not improve disruptive potency (e.g., A → B; Fig. 2a), leading to variants that lie to the upper left of the parental efficiency line. In contrast, variants with improved disruptive efficiency would lie to the lower right of the original efficiency line, with better disruptive potency for a given affinity (A → D; Fig. 2a). Isolating more and less efficient variants could provide insights into sequence, kinetic, and structural features that influence the disruptive mechanism in a given system.

To establish a yeast-based high throughput screening method for selecting more efficient disruptive inhibitors, we considered three potential selection schemes. The first scheme, pure affinity maturation, does not provide any direct selection pressure for disruptive potency or efficiency (Fig. 2b). A second scheme selects variants from a library exposed to ligand:receptor complexes for a fraction of the complex half-life to identify variants with improved disruptive potency and speed (Fig. 2c), as they compete by displacing the receptor. However, this scheme does not control for changes in the disruptive inhibitor binding affinity for the ligand and therefore does not directly select for disruptive efficiency. We therefore devised a third scheme that includes co-staining with a mutant ligand that binds the inhibitor but not the receptor (Fig. 2d). In this scheme, the inactive ligand is titrated to a sub-saturating concentration and stains yeast according to the inhibitor ligand binding affinity, while ligand:receptor complexes report on the disruptive activity of the inhibitor. This dual staining reflects disruption and affinity and could therefore enable direct selection of more efficient ($ID_{50}/K_D$) variants. Here we used a biotin-labeled $IgE-Fc_{2-4}$ as the ligand, a soluble FcεRIα-Ova fusion for the receptor, and a non-FcεRIα binding $IgE-Fc_{3-4}$ construct ($G335C-IgE-Fc_{3-4}$) as the inactive ligand mutant (Fig. 2e)[18]. We have previously shown that omalizumab and E2_79 bind with similar affinity to WT and G335C mutant $IgE-Fc_{3-4}$ fragments and both have an epitope confined to the stable Cε3 domain[3,8]. Nevertheless it is important to consider that the conformational state of the inactive ligand mutant could favor the selection of antibodies with unanticipated conformational bias.

To validate these staining schemes, we displayed three anti-IgE molecules on *S. cerevisiae* a weak disruptor (omalizumab scFv), an efficient disruptor (E2_79), and a non-FcεRIα-competing DARPin (E3_53) (Fig. 2f). Each control reagent binds titrations of free IgE or preformed complexes of IgE:FcεRI as expected (Fig. 2g). Staining with the IgE:FcεRI complex alone provided some discrimination of each displayed anti-IgE, however use of the disruptive efficiency stain with IgE:FcεRI complex and the non-FcεRIα binding $G335C-IgE-Fc_{3-4}$ significantly improved discrimination of omalizumab and E2_79 (Fig. 2h).

Disruptive inhibitors must bind to intact ligand-receptor complexes briefly prior to disruption. Therefore it is possible

that during selections with intact complexes, variants that stabilize complexes or bind complexes non-competitively could also be enriched. Therefore a final "co-binding" assay with the ovalbumin-tagged FcεRIα (FcεRIα-Ova) and a FITC labeled anti-Ovalbumin antibody was used to further discriminate non-competitive binders (E3_53) from competitive and disruptive inhibitors (E2_79 and omalizumab) (Fig. 2i–j).

Using the efficiency stain, we selected omalizumab variants with improved disruptive efficiency and potency. We constructed a series of Error-Prone PCR (EP) omalizumab libraries, and then shuffled the mutations from EP library hits using the staggered extension process (StEP) to generate StEP libraries prior to further rounds of selection[19]. All selections were conducted on four libraries named for the mutation technique (EP of StEP) and generation (1 or 2) (Fig. 3a Supplementary Fig. 2).

The first error-prone library (EP1) progressed during four rounds of selection (Fig. 3b), and clones showed improvement by the disruptive efficiency screen (Supplementary Fig. 2b). Subsequent iterations of selections using EP libraries and shuffling were then employed to further improve variants (Supplementary Fig. 2c), and a combination of sequencing data and the efficiency screens isolated improved clones (Supplementary Fig. 2c, d). Clone C02 was selected for further characterization alongside clone 813 as an intermediate comparator.

**Omalizumab variants exhibit enhanced affinity and disruptive efficiency.** Cloned hits and controls were retransformed into fresh yeast, induced, and stained with the disruptive efficiency stain followed by fixation to reduce sample to sample variation caused by dissociation of bound IgE during staining and data acquisition. In this assay Clone 813 and C02 outperform omalizumab scFv, and C02 appears similar to E2_79 (Fig. 3c). During selections the library (and clone C02) appeared to outperform E2_79 (Supplementary Fig. 2c), however during these experiments fixation was not employed.

Despite selections with intact IgE:FcεRIα complexes, clone 813 bound intact complexes to a similar extent as omalizumab scFv in co-binding assays (Fig. 3d). In contrast C02 showed a modest enhancement of co-binding (Fig. 3d). In multicolor experiments staining for both bound IgE and FcεRIα receptor, each omalizumab variant showed a degree of co-binding signal proportional to the IgE binding signal in 2D plots, suggesting that co-binding can occur at low levels prior to omalizumab mediated displacement (Fig. 3e), consistent with prior reports that omalizumab inefficiently binds intact complexes[15].

As a control for affinity maturation without explicit selection for disruptive efficiency we also produced a high-affinity omalizumab variant (HAE) previously engineered by phage display and taken into clinical trials[20]. When displayed on yeast the HAE scFv exhibits a weaker disruptive efficiency signal than C02 and looks comparable to 813 in terms of efficiency (Fig. 3f). The HAE control also showed enhanced co-binding to intact $bIgE-Fc_{2-4}$:FcεRIα-Ova complexes (Fig. 3g).

To evaluate the success of the yeast-based selections and screening assays purified scFvs were produced (Supplementary Fig. 3a–b) and used in binding and disruption studies. Compared to omalizumab, clone 813 showed a modest improvement in affinity driven primarily by slower dissociation, an almost two-fold improvement in $ID_{50}$, and a small increase in disruptive efficiency (Fig. 3h and Supplementary Fig. 3c–d, Supplementary Tables 3 and 4). Clone C02 showed a fourfold improvement of affinity with slower dissociation, a tenfold improvement in $ID_{50}$, and a five-fold improvement in efficiency, making it almost as efficient as the DARPin E2_79 (Fig. 3h and Supplementary Fig. 3c–d, Supplementary Tables 3 and 4). In contrast HAE scFv

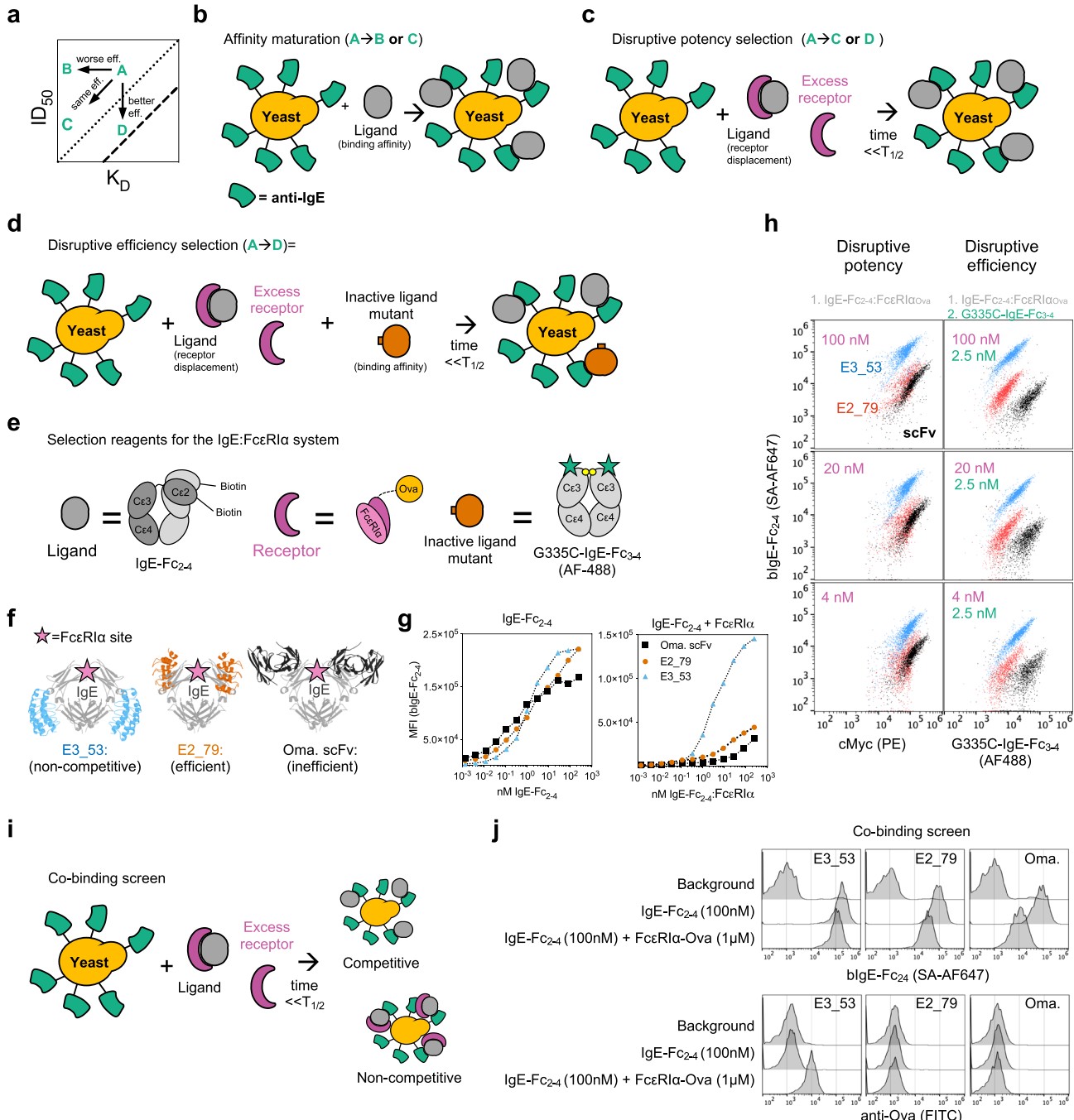

**Fig. 2 Directed evolution of omalizumab variants using a disruptive efficiency screen. a** Schematic plot of $ID_{50}$ vs $K_D$, with two different efficiency trend lines plotted in dotted or dashed lines. Hypothetical changes in the behavior of a disruptive inhibitor "A," are diagramed to show changes in affinity only (A → B), improvement in affinity and disruptive potency (A → C), or pure improvement in disruptive efficiency (A → D). **b** Cartoon schematic of standard affinity maturation selection approaches. **c** Cartoon schematic of a disruptive potency selection pathway. **d** Cartoon schematic of a disruptive efficiency selection pathway. **e** Schematic with recombinant constructs used to engineer the IgE:FcεRIα complex. **f** Structure of displayed anti-IgE agents bound to IgE-Fc$_{3-4}$ with FcεRIα binding site highlighted. **g** Yeast-displayed anti-IgE agents stained with a titration of IgE-Fc$_{2-4}$ (left) or IgE-Fc$_{2-4}$:FcεRIα (right). **h** (Left) singlet-cMyc+ anti-IgE yeast stained for disruptive potency selection and anti-cMyc at indicated concentrations. (Right) singlet-cMyc+ yeast stained for disruptive efficiency selection at indicated concentrations. **i** Schematic of co-binding screen. **j** Normalized histogram of singlet-cMyc+ yeast anti-IgE controls stained with secondary reagents alone to assess background signal on yeast, binding to bIgE-Fc2–4 alone (100 nM), or co-binding to bIgE-Fc2–4 (100 nM) in complex with FcεRIα-Ova (1 µM). Surface bound bIgE-Fc$_{2-4}$ was detected with SA-647 (top) and surface bound FcεRIα-Ova was detected with anti-Ova FITC (bottom). Source data are provided as a Source Data file.

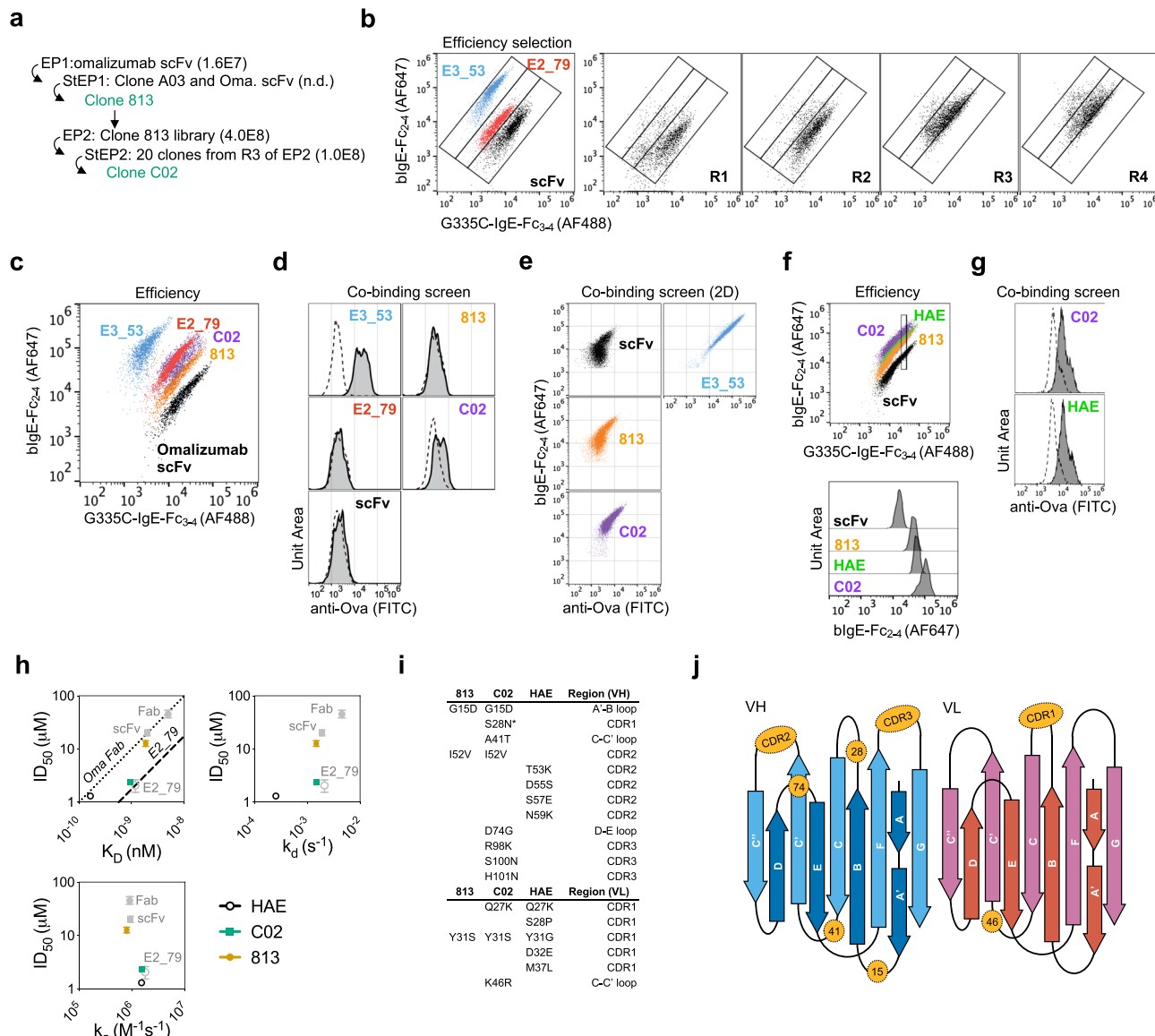

**Fig. 3 Omalizumab variants exhibit enhanced affinity and disruptive efficiency. a** Overview of selections with number of transformants per library in brackets. Full details of selections outlined in Supplementary Fig. 2. **b** Controls and freshly induced R1–4 of EP1 library stained with disruption efficiency stain. **c** Hits and controls were stained with disruption efficiency stain, followed by fixation. **d** Histogram of yeast hits and controls stained for co-binding:100 nM bIgE-Fc$_{2-4}$ (dotted line) or precomplexed bIgE-Fc$_{2-4}$:FcεRIα-Ova (100 nM:1 μM-solid line), with fixation. **e** Two color co-binding screen of omalizumab variants stained with bIgE-Fc$_{2-4}$:FcεRIα-Ova (100 nM:1 μM) to assess correlation between bIgE-Fc$_{2-4}$ and FcεRIα-Ova signals as compared to a non-competitive control (E3_53). **f** HAE yeast stained with disruption efficiency stain with fixation compared to omalizumab variants. For clarity a fraction of G335C-IgE-Fc$_{3-4}$ positive cells were gated by their G335C-IgE-Fc$_{3-4}$ staining intensity, and the relative intensity of biotin-IgE-Fc$_{2-4}$ was displayed by histogram. **g** C02 and HAE stained with co-binding stain with fixation. **h** ID$_{50}$ with 95% CI from fits in Supplementary Fig. 3d vs. K$_D$, k$_a$, or k$_d$ by variant. The efficiency ratio for omalizumab Fab and E2_79 are plotted as benchmarks for other variants (dotted and dashed line). **i** Amino acid mutations in clones 813, C02, and HAE relative to omalizumab. **j** Distribution of mutations in (i) mapped onto topology of VH and VL domains.

dissociated extremely slowly, bound IgE more than 5-fold tighter than C02, yet was less than twice as disruptive and thus exhibited a disruptive efficiency more similar to the parental omalizumab and clone 813 (Fig. 3h and Supplementary Fig. 3c–d, Supplementary Tables 3 and 4). These observations match the ranking predicted in yeast-based assays and C02 and HAE demonstrate that binding kinetics and disruptive potency can be engineered independently. Consistent with their divergent disruptive profiles, C02 and HAE contain two mutated VL CDR1 residues in common (Q27K, Y31S/G), but no overlapping VH mutations, highlighting that each antibody improved its affinity and disruptive potency via distinct paths (Fig. 3i–j).

## Structural analysis of high-affinity disruptive omalizumab variants.

To identify structural correlates of disruptive potency and efficiency we crystallized native omalizumab scFv, clone 813, clone C02, and clone HAE in complex with a smaller IgE Fc fragment, IgE-Fc$_{3-4}$, which contains the full omalizumab epitope (Supplementary Table 5). Prior structural studies suggested that the degree of potential steric overlap between inhibitor and FcεRIα correlated with their disruptive-potency and efficiency[21]. We therefore aligned all antibody:IgE pairs relative to the Cε3 domain to analyze conformational arrangements that could impact steric overlap with FcεRIα. The binding pose of omalizumab is well conserved in CDR proximal regions in all

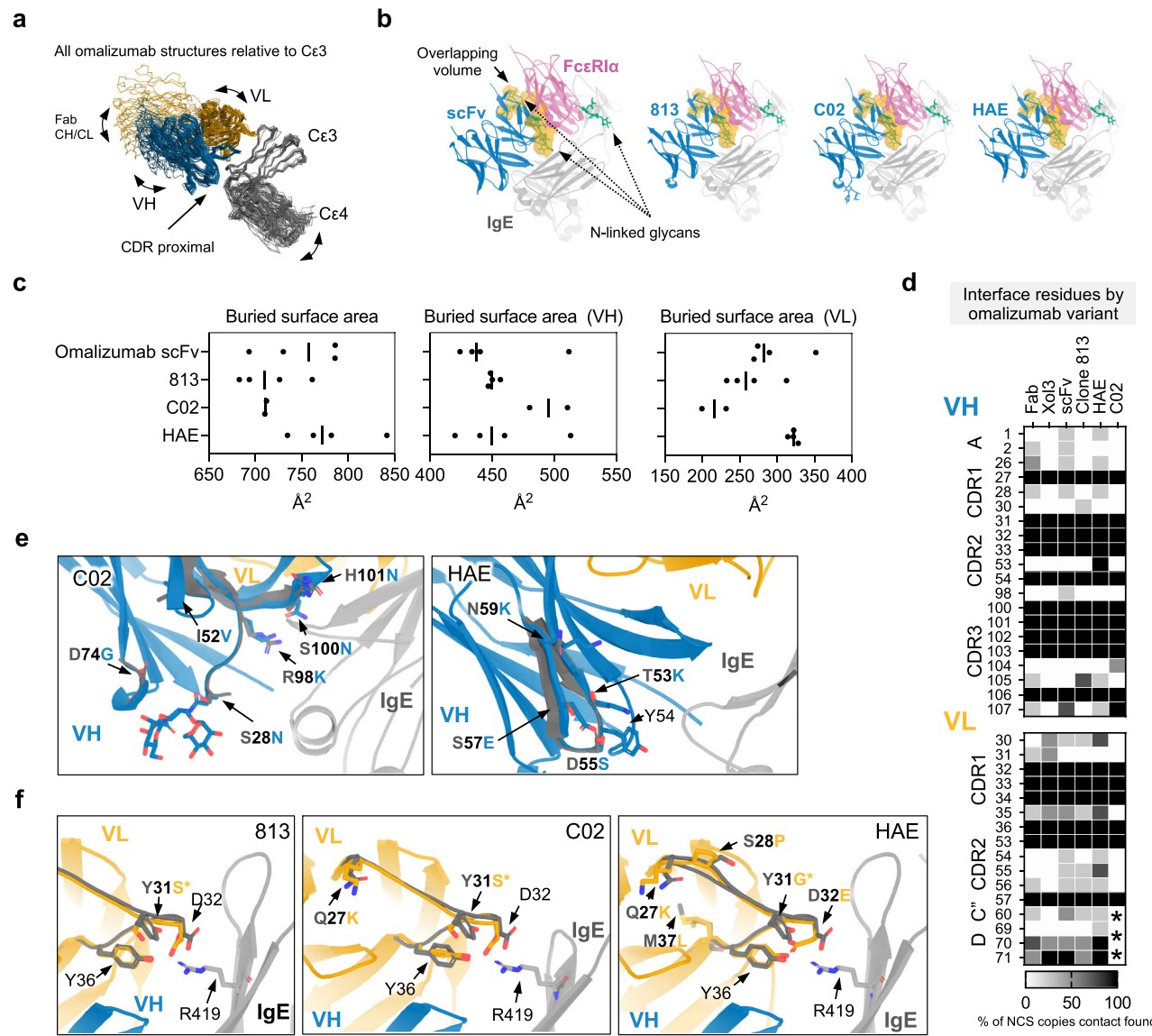

**Fig. 4 Structural analysis of high-affinity disruptive omalizumab variants. a** Ribbon diagram of all omalizumab variants and omalizumab:IgE complexes aligned relative to Cε3 showing conservation of CDR proximal regions and variation in Fab CH/CL, IgE Cε4, VH and VL domains. **b** scFv structures were aligned relative to Cε3 domain at site 2 of the asymmetric FcεRIα:IgE-Fc complex. Maps of FcεRIα and each scFv were generated from each model and the volume of steric overlap between each map is depicted in orange, with FcεRIα associated glycans shown in green. **c** Plots of the buried surface area of all NCS related copies of each scFv, VH, and VL with IgE with mean surface area indicated. **d** Heat map of omalizumab contacts at interfaces across all structures. Shading reflects percentage of NCS related IgE:omalizumab interfaces at which a contact was identified. Asterisks denote region in which lack of density precluded modeling/interface-identification. **e** Detailed views of VH mutations (blue) aligned by variant to the native omalizumab scFv (dark gray), with IgE (light gray). **f** Same as in **c** for VL mutations, with IgE-R419 highlighted for reference. Source data are provided as a Source Data file.

structures, but conformational flexibility is present in regions distal to the binding interface (Fig. 4a). Despite the flexibility across crystal structures we observed no correlation between disruptive potency and the binding pose of individual VH or VL domains, or the combined VH + VL structures relative to the bound IgE Cε3 domain. (Supplementary Fig. 4a). Consistent with this observation the predicted steric overlap with receptor for each scFv in structural alignments was extremely similar (Fig. 4b).

Conformational rearrangements within the IgE-Fc have been extensively studied and are well known to effect receptor binding[22–25]. Therefore we also assessed if scFv variants could modulate "open," and "closed," conformations of the IgE-Fc[22], as these conformation rearrangements are known to modulate IgE:FcεRIα interactions. This analysis revealed no correlation between the Fc conformational states and disruptive potency

(Supplementary Fig. 4b, c). Rearrangements of the Cε2 domains are also known to regulate receptor binding, however these domains are not present in these structures, and we cannot exclude the possibility that these scFvs differentially interact with or alter the orientation of these domains.

The omalizumab IgE footprint was also well conserved across structures, yet the buried surface area of C02 contained a relatively larger VH and smaller VL footprint (Fig. 4c, d). Minor changes were observed near epitope proximal mutations, across scFv copies related by non-crystallographic symmetry (NCS), or in regions adjacent to solvent exposed side chains that could not be modeled (Fig. 4d). Although the antibody footprint was similar, there were substantial differences in the distribution of interface and interface adjacent mutations. C02 accumulated mutations within the heavy chain HCDR3 and forms polar

contacts via the H101N mutation and intramolecular hydrogen bonds within the HCDR3 loop via the S100N and R98K mutations (Fig. 4e). C02 also acquired mutations in the HCDR1 loop which introduced a N-linked glycan proximal to the interface, yet the sugar does not appear to form stable contacts with IgE or produce a dramatic reorganization of adjacent residues (Fig. 4e). In contrast, HAE contains a cluster of different mutations within HCDR2 and only one extends to the omalizumab interface (T53K) (Fig. 4e). The remaining HAE HCDR2 mutations are distal to the interface and form intrachain contacts, but surround Y54, a key interface residue found in all omalizumab:IgE complexes.

Within the VL domain, 813, C02, and HAE converged on a Y31S or G substitution that reduces steric bulk proximal to residue 32 (Fig. 4f). This substitution facilitates the formation of intrachain hydrogen bonds in HAE and C02 at VL position D/E 32 and Y36 which form the posterior edge of the binding pocket for R419 of IgE, a residue shown to be critical for omalizumab binding (Supplementary Fig. 4d)[7]. Adjacent to these core mutations, C02 and HAE both converged on an upstream Q27K mutation at the base of the LCDR1, and HAE contains additional mutations in and proximal to the LCDR1 (S28P, M37L) which are not directly in contact with IgE (Fig. 4f). These data demonstrate that alterations that stabilize the LCDR1 loop are critical to affinity and disruptive potency enhancement in two omalizumab variants isolated through distinct selection methods.

Together these studies highlight shared LCDR1 mutations that enhance affinity and demonstrate that small changes to the VH and VL binding footprints and polar bonds help distinguish the efficient disruptive omalizumab variant C02 from HAE, 813, and omalizumab scFv. However, these IgE bound structures do not reveal the relevance of these mutations during disruption. We therefore reasoned that the effect of each mutant might be revealed in structures of the predicted trimolecular omalizumab:IgE:FcεRI complex along the disruption pathway.

**A disulfide "locked" IgE-Fc2–4:FcεRIα complex enables the selection of omalizumab variants that trap a disruption intermediate.** To study transient intermediates along the disruption pathway, we designed a disulfide engineered IgE-Fc$_{2-4}$:FcεRIα "locked" complex. The covalently stabilized locked complex should be resistant to full dissociation and allow trapping of partially disrupted intermediates for structural studies. A favorable disulfide bond was predicted between IgE residue G335 and FcεRIα residue W156[26]. In the smaller IgE-Fc$_{3-4}$ fragment the G335C mutation forms an interchain disulfide bond and traps the IgE-Fc in a closed conformation, but in the IgE-Fc$_{2-4}$ fragment the distance between G335 residues is larger and constrained by adjacent Cε2 domains. We therefore hypothesized that when co-expressed with the FcεRIα mutant W156C a fraction of IgE-Fc$_{2-4}$ mutant G335C might form a intermolecular IgE:FcεRIα disulfide bond, covalently linking the two proteins proximal to site-two of the asymmetric IgE:FcεRIα complex (Fig. 5a).

Using a two-step affinity and ion-exchange purification scheme a homogenous complex containing a His-tagged FcεRIα and unlabeled IgE-Fc$_{2-4}$ was isolated (Supplementary Fig. 5a, b). This species was stable in SDS, yet reducible to monomeric components in the presence of DTT (Supplementary Fig. 5c, d). The locked complex exhibited binding characteristics similar to wild type IgE-Fc$_{2-4}$:FcεRIα complexes with robust binding to the non-competitive inhibitor E3_53 and weak binding to omalizumab scFv, Fab, and E2_79 at concentrations ~50–100 fold higher than their respective K$_D$ for free IgE-Fc$_{2-4}$ (Fig. 5b). We also assessed the binding affinity of yeast-displayed anti-IgE agents for

biotinylated locked-complex as compared to free bIgE-Fc$_{2-4}$. In agreement with biolayer interferometry (BLI) studies E3_53 displays similar binding affinity for both the locked-complex and free-IgE, while omalizumab scFv binds free-IgE well, but not the locked-complex (Fig. 5c). The disruptive variants C02 and HAE display an intermediate profile with robust free-IgE binding and enhanced locked-complex binding as compared to omalizumab scFv, further indicating that the locked complex can segregate agents by their disruptive potency (Fig. 5d).

To improve the binding of scFv variants to the locked complex for structural studies, we produced a small shuffled library of C02 and HAE. After five rounds of selection the majority of clones converged on variants employing C02-VH and HAE-VL sequences with sporadic additional mutations (Supplementary Fig. 5e–f), yet clones with the tightest locked-complex binding in yeast (Fig. 5e) shared a novel mutation distal to the binding interface in the E-strand of the VL chain (clone A4:D74G, clone 7:D74H, and clone 16:D74Y) (Fig. 5f and Supplementary Fig. 5f). Unlike the other clones, clone 7 was identical to C02 beyond the D74H mutation, which was independently identified in early selections (clone F04, D74H Supplementary Fig. 2b).

Confirmatory BLI binding studies with scFvs and the locked complex could not be fit by 1:1 binding models, although we had previously hypothesized that only a single epitope would be accessible during the disruption of IgE:FcεRIα complexes. Both C02 and HAE appear to have one high and one low affinity binding site within the locked-complex (56.9 nM–113 nM and 2.86 nM–25.8 nM respectively) (Fig. 5g, Supplementary Table 6). The D74H mutation increases the binding affinity of clone 7 compared to C02 (Fig. 5g) and binding models for all clones with the VL E-strand mutations (A4, 7, and 16) estimate extremely slow dissociation rates in a subset of binding events (Supplementary Fig. 5g, Supplementary Table 6). These measurements indicate that disruptive variants can bind IgE:FcεRIα complexes at low concentrations prior to disruption, and that non-equivalent omalizumab epitopes exist in the intact IgE:FcεRIα complex.

We then tested each clone in yeast for receptor co-binding using wildtype bIgE-Fc$_{2-4}$:FcεRIα-ova complexes. While both clones A4 and 16 showed a modest increase in co-binding of FcεRIα-ova compared to C02 or HAE, clone 7 displayed significantly greater co-binding with a phenotype intermediate to C02 and the non-competitive binder E3_53 (Fig. 5h). Consistent with these observations soluble scFvs of clone A4 and 16 showed a disruptive ID$_{50}$ similar to C02 or HAE, while the ID$_{50}$ of clone 7 increased five-fold relative to C02 despite displaying a higher affinity for free-IgE (Fig. 5i, Supplementary Fig. 5g, h, and Supplementary Tables 7 and 8). The resulting disruption efficiency of clone 7 is lower than any other omalizumab variant and this decrease in efficiency is caused by a single point mutation outside the antigen binding site. These results suggests that clone 7 not only binds tightly to the locked complex but also stabilizes an intermediate state along the disruption pathway in native protein complexes.

**Cryo-EM structure of a partially disrupted antibody:IgE:FcεRI complex.** We selected clone 7 for further structural studies given that it varied from C02 by a single amino acid, was capable of disruption, and appeared to stabilize intact complexes in wildtype IgE:FcεRIα binding studies. Stable clone 7:IgE-Fc$_{2-4}$(G335C):FcεRIα(W156C) complexes could be detected by SEC (Supplementary Fig. 6a) and the structure of the clone 7-scFv$_2$:IgE-Fc$_{2-4}$(G335C):FcεRIα(W156C) complex was determined by single particle cryo-EM to a nominal resolution of ~7.3 Å (Supplementary Table 9, Fig. 6a) with regions proximal to antibody and receptor

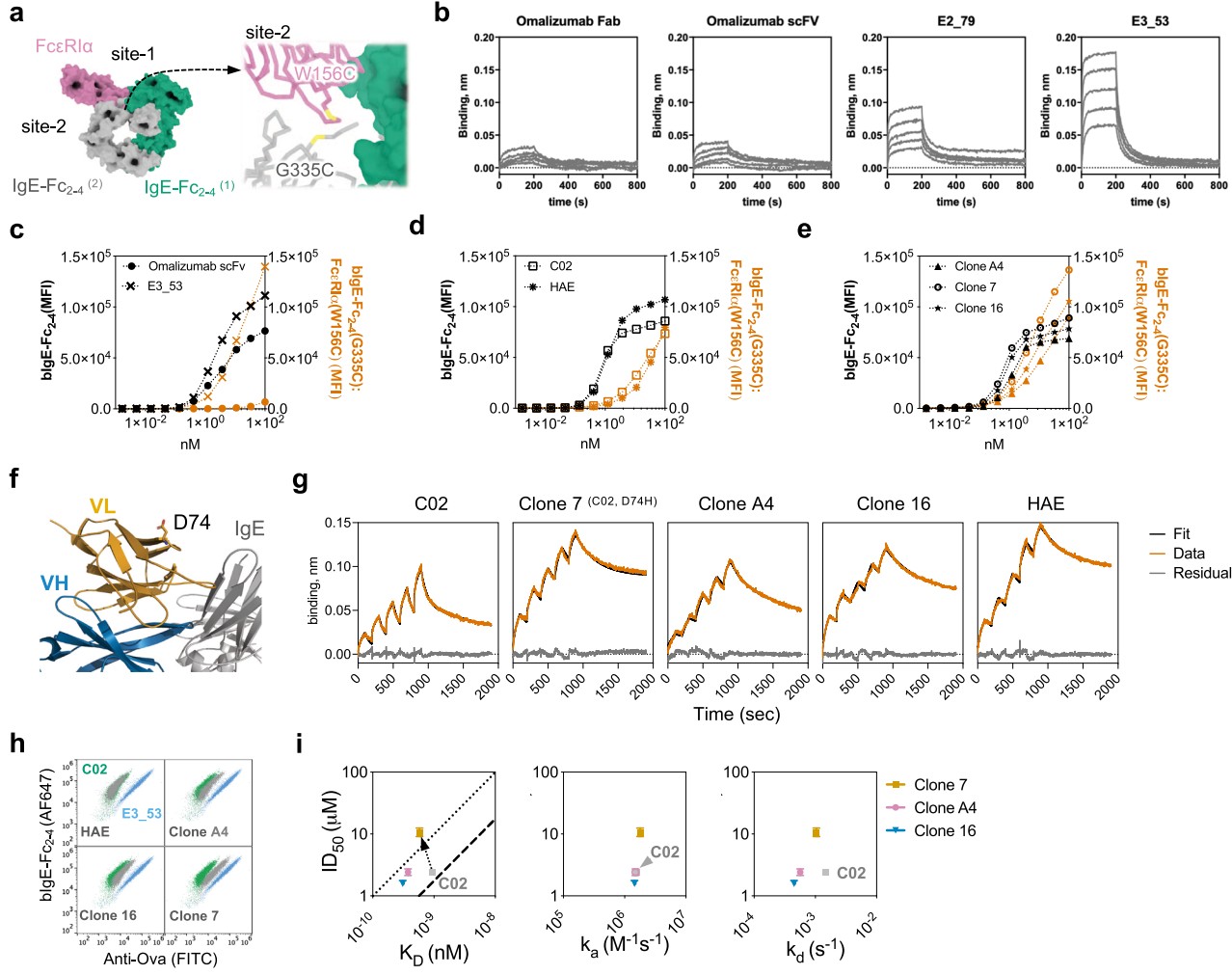

**Fig. 5 A disulfide "locked" IgE-Fc$_{2-4}$:FcεRIα complex enables the selection of omalizumab variants that trap a disruption intermediate. a** IgE-Fc$_{2-4}$:FcεRIα complex (2Y7Q), with the interface of FcεRIα and IgE at "site-2" revealed to display position of IgE-Fc$_{2-4}$ (G335C) and FcεRIα (W156C) mutations. **b** Octet binding studies with locked-complex on tips exposed to anti-IgE agents as indicated in a twofold serial dilution from 400 nM to 25 nM. **c** Binding profiles of yeast-displayed omalizumab scFv and E3_53 to free bIgE-Fc$_{2-4}$ (left axis, black) or locked-complex (right axis, red). **d** Same as in **c** for yeast-displayed C02 and HAE. **e** Same as in **c** for clones A4, 7, 16. **f** Schematic of novel E-stand mutation position relative to omalizumab:IgE complex structure. **g** BLI SCK binding studies of anti-IgE variants as indicated with locked-complex on tips with anti-IgE agents in a twofold serial dilution from 1600 nM to 100 nM. **h** Yeast-displayed omalizumab clones 7, A4, and 16 stained with bIgE-Fc2-4 (100 nM):FcεRIα-Ova (1 μM) complexes relative to yeast expressing C02, HAE, and E3_53 (samples fixed). **i** ID$_{50}$ with 95% CI from fits in Supplementary Fig. 5h vs. K$_D$, k$_a$, k$_d$ for A4, 7, and 16. The efficiency ratio for omalizumab Fab and E2_79 are plotted as benchmarks for variants (dotted and dashed line). Source data are provided as a Source Data file.

interfaces resolved at higher local resolution (<7 Å) (Supplementary Fig. 6c–e). Although size exclusion chromatography suggested a binding stoichiometry between 1 and 2 scFvs per complex, no particle classes containing only one scFv were identified, consistent with the complex binding behavior observed in BLI studies. Density for all domains within the complex was well resolved and even allowed modeling of the core IgE-Fc-glycans at N394 (Fig. 6a), glycans on FcεRIα, and glycans on each scFv (Supplementary Fig. 6g), but was notably worse for Cε2 which had the poorest fit to the density (Supplementary Fig. 6f)

In this disruption-intermediate, clone 7 scFvs occupy both site-1 and site-2 epitopes of the IgE-Fc homodimer in a pseudosymmetric manner. One of the scFvs is oriented with its VL domain adjacent to the IgE-Cε2 domains (site-1), while the other scFv VL domain is adjacent to FcεRIα (site-2). Both of these novel VL interfaces involve the unique D74H mutation in the E-strand of clone 7 (Fig. 6b), providing a structural explanation for the increased binding affinity of clone 7 to the locked complex and enhanced co-binding of wildtype IgE:FcεRIα complexes.

In their native binding poses on IgE, omalizumab and FcεRIα would physically overlap each other, suggesting that one or both molecules must be displaced in this intermediate state. Within the antibody, we observe a distortion of the scFv VH/VL conformation relative to the native-C02 structure. Although the low resolution of the EM structure precludes detailed models of these conformational rearrangements, the density and alignment to existing structures supports the following conclusions: 1. The VH domain binding pose remains relatively unperturbed 2. The VL-CDR1 loop, and the site of the most convergent scFv mutations is displaced from the free-IgE binding pose at both sites (Supplementary Fig. 6h–j). Therefore, the adjacent FcεRIα and Cε2-domains must partially block omalizumab binding and impose distortions on the VH/VL conformation.

Despite rearrangements of the scFv, alignment of the site-2 scFv to the native IgE:FcεRIα structure by the proximal IgE-Cε3 domain demonstrates that steric clashes between the native FcεRIα position would persist (Fig. 6c), indicating that FcεRIα displacement was required for clone 7 binding. We measured

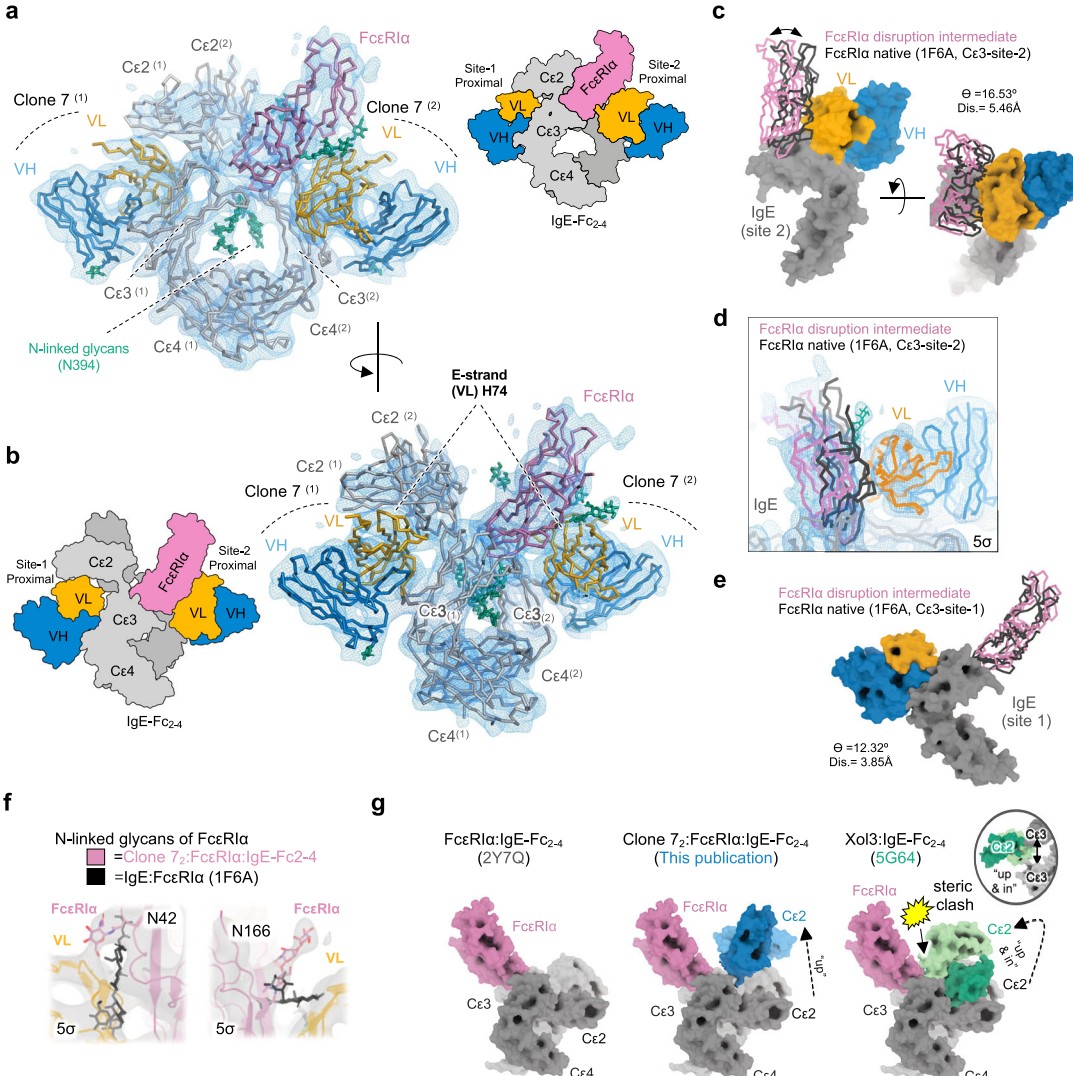

**Fig. 6 Cryo-EM structure of a partially disrupted IgE:FcεRI complex. a** Front view of model fit to cryo-EM density map contoured at 5σ, with cartoon schematic (right) **b** Side view of model fit to cryo-EM density map contoured at 5σ, with cartoon schematic (left). **c** Displacement of FcεRIα in disruption-intermediate (magenta) as compared to the native IgE:FcεRIα structure (black) relative to the site-2 FcεRIα binding site. Quantification of angle (Θ) and distance (Å) of displacement indicated. **d** Detailed view of the cryo-EM density map at the site-2 FcεRIα binding site. **e** Displacement of FcεRIα in disruption-intermediate (magenta) as compared to the native IgE:FcεRIα structure (black) relative to the site-1 FcεRIα binding site. Quantification of angle (Θ) and distance (Å) of displacement indicated. **f** Relative positions of FcεRIα glycans compared to native IgE:FcεRIα structure with density contoured at 5σ. **g** Conformations of Cε2 domains from disruption-intermediate (this publication) and omalizumab Fab Xol3:IgE-Fc$_{2-4}$ (5G64) relative to the native site-1 FcεRIα binding pose (2Y7Q). Yellow accent denotes region of steric clash between Cε2 and FcεRIα, with inset depicting back view of relative Cε2/3 positions in 5G64.

FcεRIα displacement from the native binding pose at site-2 or site-1, by aligning the disruption-intermediate structure to the native IgE:FcεRIα structure through the corresponding Cε3 domains (Fig. 6c–e). After alignment we visualized the relative displacement of FcεRIα across models and quantified the displacement using *angle_between_domains* from Pymol Script Collection (PSICO). Although the low resolution limits a detailed analysis of this alternate conformation, it is clear that FcεRIα undergoes substantial rearrangements to fit large portions of the main chain into the experimental density (Fig. 6d). The entire FcεRIα chain is displaced relative to both Cε3 binding sites in the disruption-intermediate, with more pronounced displacement at site-2 (θ = 16.5˚, displacement = 5.5 Å), where the steric conflicts between the clone 7 scFv and FcεRIα need to be resolved (Movie 1). Additional predicted site-2 clashes occur between FcεRIα N-linked glycans (N42 and N166), which retain similar

orientations across all prior IgE:FcεRIα and FcεRIα structures[3,21]. Notably, in the disruption-intermediate, density at these glycan sites suggests that they reorient and adopt distinct conformations to accommodate clone 7 binding (Fig. 6f).

Adjacent to the site 1 proximal omalizumab epitope, distal to FcεRIα, the Cε2 domains adopt a conformation not seen in prior IgE-Fc structures to accommodate antibody binding (Supplementary Table 10). Alignment of the disruption-intermediate complex to site-1 of the native IgE:FcεRIα structure reveals that this Cε2 conformation could accommodate the simultaneous binding of FcεRIα and omalizumab without overlap of Cε2 and FcεRIα (Fig. 6g). This is distinct from the Cε2 conformation observed in omalizumab(Xol3):IgE-Fc$_{2-4}$ structure, where the Cε2 domains pack between the tips of the extremely open Cε3 domains, overlapping the native FcεRIα position[15] (Fig. 6g). This observation suggests that non-disruptive omalizumab binding

could occur at site 1, although antibody binding would block Cε2 dimer interactions with Cε3 and Cε4 domains that are known to stabilize the complex[23].

**Correlates of potency and safety in allergic effector cell desensitization.** The multiple binding sites observed in the locked complex structure suggests that binding events during antibody-mediated disruption could drive receptor crosslinking and spontaneous activation (anaphylactogenicity). We therefore produced full IgG1 antibodies of the most efficient disruptive variant (C02), a high-affinity variant (HAE), and a variant that was potently disruptive but bound tightly to the locked complex (clone 16). We then tested each antibodies' ability to spontaneously activate mouse bone marrow-derived mast cells transgenic for human FcεRIα (BMMCs[tg]) and isolated primary human basophils (Supplementary Fig. 7a–f). Given the role of VH/VL flexibility in the $ID_{50}$ and affinity of omalizumab, we also produced H2L2-IgG1 constructs with flexible Gly-Gly linkers at each Fab elbow for C02 and HAE variants to better reflect the flexibility of scFvs used for selections. The IgG constructs displayed a similar trend to scFv variants in bead-based disruption assays, yet the C02_H2L2_IgG variant was modestly improved and showed similar potency to HAE_IgG (Fig. 7a). Similar to omalizumab, most variants did not spontaneously activate either BMMCs[tg] or human basophils. However, despite showing robust disruption in bead-based assays, clone 16 induced dose-dependent activation of BMMCs[tg] and human basophils (Fig. 7b–c). Notably the modification of HAE_IgG to the HAE_H2L2_IgG format also led to activation in some basophil donors (Fig. 7c). Given the similar disruption profiles of HAE_IgG to the HAE_H2L2_IgG in bead-based disruption experiments, this data suggests that even small changes in the balance of binding and disruption events at the cell surface may favor receptor crosslinking and promote cell activation. These data suggest that HAE rests on the verge of cell activation, consistent with sporadic hypersensitivity events that terminated phase 2 clinical trials[27] and warrants further investigation in more subjects.

Given that some clones were anaphylactogenic, we measured IgG binding to intact IgE:FcεRIα complexes by BLI (Fig. 7d). In these experiments, a loss in signal can occur following removal of IgE from IgE:FcεRIα complexes. However, simultaneous non-disruptive antibody binding events could produce net-positive or net-neutral BLI binding signals. Strikingly, both anaphylactogenic and non-anaphylactogenic variants show pronounced complex formation over a short timeframe, while omalizumab shows little detectable association. A control off target anti-CoV2 antibody (D10) showed no significant association or dissociation of IgE:FcεRIα complexes on the tip in the same assay, while the non-competitive DARPin E3_53 formed stable complexes with IgE:FcεRIα complexes (Supplementary Fig. 7g). In these BLI assays the stability of the inhibitor binding correlates well with the spontaneous activation profile in BMMCs[tg] and human basophils. In particular, the most anaphylactogenic variant, clone 16, has the most prolonged net-positive signal and HAE variants show an intermediate profile. These results suggest that the balance of disruptive versus non-disruptive binding events, and in particular the dwell time on receptor complexes, are critical safety correlates for non-activating disruptive antibodies.

We then used the most potent non-anaphylactogenic variants (HAE_IgG or C02_H2L2_IgG), to assess their ability to rapidly desensitize BMMCs[tg]. Although omalizumab can accelerate IgE dissociation, this effect is minimal over the course of hours at physiological concentrations. In contrast the observed $ID_{50}$ of HAE_IgG1 and C02_H2L2_IgG1 for stripping IgE from BMMCs[tg] falls in the nanomolar range after a 6-h treatment (216 nM, 95%CI [174.2–263.2 nm] and 316 nM, 95%CI [243.4–394.1 nM]

respectively) (Fig. 7e). Furthermore, these agents are able to suppress activation of BMMCs[tg] with half-maximal inhibition in the mid nanomolar range (550.5 nM, 95%CI [425–642.7 nM] and 624.9 nM, 95%CI [412.8–805.4 nM] respectively) (Fig. 7f).

We also isolated primary human basophils from three grass-allergic donors to measure the inhibitory profile of each antibody in human cells. These experiments confirmed that both agents were capable of completely removing cell surface IgE and suppressing IgE dependent activation in the mid nanomolar range (Fig. 7g, h). Interestingly, C02_H2L2 was consistently more potent than HAE in basophil stripping and inhibition experiments although both exhibited similar function in BMMCs[tg]. Furthermore, both HAE and C02_H2L2 significantly desensitized cells at a concentration of 500 nM in 6 h (Fig. 7i). In comparison, omalizumab had little effect on basophil IgE levels and signaling even at the highest concentrations studied.

**Discussion**

Stable protein-protein complexes are central to the pathophysiology of many human diseases, but the therapeutic targeting of such complexes remains largely unexplored. Antibodies and antibody alternatives that are specifically selected to act via facilitated dissociation could rapidly dissociate such complexes to accelerate therapeutic effects. We have demonstrated a pathway to engineer facilitated dissociation and disruptive efficiency of macromolecular agents using soluble receptor, ligand, and an inactive ligand mutant. In this manuscript we have not directly isolated novel disruptors from a library of binders, yet we can classify a range of anti-IgE agents with distinct epitopes, affinities, and disruptive potencies suggesting the techniques could be used for de novo selections. Furthermore, these reagents have been described for many receptor-ligand pairs, such as CD47 and SIRPα[28], and antibodies that can disrupt CD47:SIRPα complexes have already been described[5]. We anticipate that multiple protein systems should be amenable to the isolation and engineering of disruptive inhibitors, although the importance of disruptive efficiency and dwell time of disruptive molecules will depend on the underlying biology of the system. In the context of IgE:FcεRIα complexes and allergy, we show that the efficiency of disruption correlates with the dwell time of disruptors. Therefore modulating efficiency specifically, and not just potency alone, may have functional consequences in other protein systems. In the case of IgE, efficient disruptors that bind with lower affinity relative to their ID50 rapidly disrupt complexes or rapidly bind and dissociate from complexes without disrupting them. In contrast higher affinity less efficient disruptors bind longer prior to disruption or dissociation. These longer dwell times increase the probability of antibody induced crosslinking of IgE:FcεRIα complexes and unwanted spontaneous activation of basophils.

Biophysical and structural studies of engineered omalizumab suggest that facilitated dissociation can be driven by small rearrangements at protein:protein interfaces stabilized by antibody binding. This mechanism of action should be applicable to many ligand-receptor pairs and does not require large conformational rearrangements of protein domains or allosteric effects that are restricted to a specific protein complex. During disruption, we show that small rearrangements can also occur within disruptive antibodies that adopt alternate conformations and employ interactions at cryptic interfaces outside of their CDR loops to modulate disruption. These events were not captured in studies of antibody:IgE complexes but became evident in studies with intact complexes and in the holocomplex structure of IgE, FcεRIα, and omalizumab. Although these structural insights are consistent with our biochemical and biophysical studies, it is important to consider the possibility that the engineered disulfide bond used to

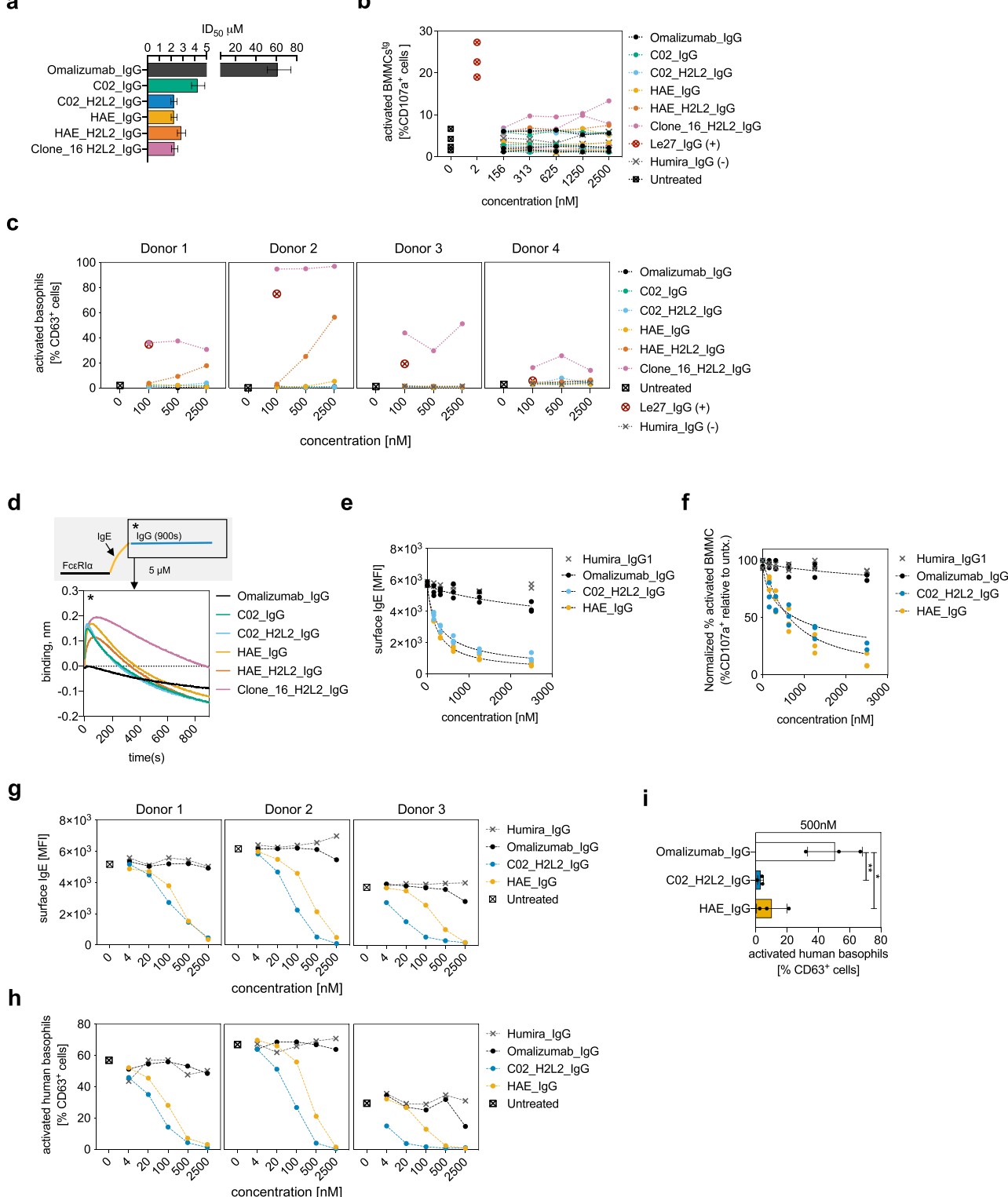

**Fig. 7 Correlates of potency and safety in allergic effector cell desensitization. a** ID$_{50}$ of omalizumab IgG variants, with 95% CI from fit of bead-based disruption assays. **b** Anaphylactogenicity of IgG variants, the activating anti-IgE antibody Le27, and off target control humira in BMMC as measured by percent CD107a$^+$ cells (individual replicates shown). **c** Anaphylactogenicity of IgG variants and controls in basophils from human allergic donors ($n = 4$) as measured by % CD63$^+$ cells (individual donors shown). **d** BLI studies of IgG binding to intact IgE:FcεRIα complexes, with schematic of assay (top). **e** BMMC inhibition assays following 6-hour treatment with IgG variants at indicated concentrations. Cultures were split and assayed for surface-IgE (**e**) or activated with NIP(7)-BSA and assayed for activation (%CD107a$^+$) (**f**). Activation data was normalized to 100% activation for untreated controls. **g** Basophil inhibition assays from grass-allergic donors following 6-h treatment with IgG variants at indicated concentrations. Cultures were split and assayed for surface IgE (**g**) or activated with 6-grass allergen mix and assayed for activation (%CD63$^+$) (**h**). **i** Mean and SD of inhibition observed by IgG variants at 500 nM after 6-hour incubation. One-way repeated measures ANOVA with Bonferroni post-hoc tests (*$p = 0.0123$, **$p = 0.0067$, $N = 3$, independent human donors). Source data are provided as a Source Data file.

trap the complex could alter the architecture of the complex itself. However, the spontaneous association of mutant IgE-Fc molecules with mutant FcεRIα receptors during co-expression, and the computational approach used to pick energetically favored disulfide bonds[26] suggest that the mutant complex is a reasonable surrogate for native complexes. Furthermore clone 7 not only bound locked complex tighter, but also stabilized native IgE:FcεRIα complexes in yeast-based studies, suggesting that the locked complex reflects a transition state observed in native complexes. Beyond the engineered disulfide, the relatively low resolution of the cryo-EM data also leaves several fundamental questions about the process of disruption unanswered, such as how CDR loops of omalizumab variants accommodate the distinct binding pose observed during disruption, or what molecular interactions occur between the antibody and complex outside of the core antibody-IgE interface. This low resolution was likely secondary to the relatively small data set and the inherent conformational flexibility of partially bound receptor and scFvs. Future studies are warranted as techniques in data acquisition rapidly improve and may facilitate a higher resolution structure determination.

Together these results not only provide a framework for understanding and engineering disruption, but they also demonstrate how the directed evolution of facilitated dissociation can have therapeutic implications. IgE mediated type-1 hypersensitivity remains a significant source of human morbidity and mortality despite our advanced understanding of the underlying molecular mechanisms. By harnessing facilitated dissociation and engineering this kinetic dimension of antibody function, we have demonstrated that we can rapidly remove IgE and desensitize human allergic effector cells using anti-IgE antibodies at concentrations observed in serum during treatment with existing anti-IgE therapeutics[29–31]. These changes allow us to completely arrest the allergic signaling cascade on timescales in which the paternal antibody omalizumab had no effect. Together with other disruptive anti-IgE inhibitors, which exhibit a range of potencies (Supplementary Table 11), these disruptive antibodies expand opportunities to significantly accelerate anti-IgE therapeutic response times in patients.

## Methods

**Protein expression and purification.** G335C-IgE-Fc$_{3–4}$ was cloned into pAcGP67A, and baculovirus was produced using the Baculogold system (Pharmingen). SF9 insect cells were infected and supernatants were harvested after 3 days, and purified with Ni-NTA affinity chromatography (Qiagen), followed by size exclusion chromatography (SEC) on a S75 10/300 GL column (GE Healthcare). The DARPin E2_79 in the pQE-30 vector was transformed into XL-1 blue *e. coli*. XL-1 *E. coli* were grown to an OD600 of 0.6 in dYT media, induced in with IPTG at 1 mM for 4–6 h at 37 °C, pelleted, and resuspended in lysis buffer (40 mM Tris-HCl, pH 7.5, 300 mM NaCl, 20 mM imidazole, 10% glycerol, and 1 mg/ml lysozyme). Resuspended cells were subjected to three freeze thaw cycles, and then sonicated with immersion sonicator. Lysates were treated with EDTA free protease inhibitor tablets (Pierce), clarified by centrifugation at 30,000 × *g* for 20 min, and filtered through 0.45uM PES filter (Millipore/Sigma). Filtered lysates were then applied Ni-NTA resin column (Qiagen) and further purified by SEC on a S200 10/300 GL column (GE Healthcare). All remaining recombinant proteins in this manuscript were cloned into pTT5 vector system and transiently expressed in HEK-6E cells[32]. In brief HEK-6E cells were grown to a density of 2.0 million cells/mL in freestyle media supplemented with G418 at 25 µg/mL and Kolliphor P188 0.1% w/v prior to transfection with precomplexed plasmid DNA (1 mg/L of cells) and 25 kDa PEI (3 mg/L-cells). After 5–6 days secreted protein was harvested following centrifugation of cells at 9000 × *g* and media clarification with a 0.45 µm PES filter (Millipore/Sigma). Proteins were purified from supernatants with affinity Ni-NTA (Qiagen) or protein-A (GenScript) chromatography before further purification by SEC on HiLoad 16/600 Superdex 200 (GE Healthcare) or Superdex 200 10/300 GL (GE Healthcare) columns.

**Protein labeling.** Proteins were labeled with Alexa Flour 488 NHS-Ester (Thermo Scientific) or Sulfo-NHS-LC-Biotin (Thermo Scientific) in PBS pH 7.4 in a 20:1 label:protein molar excess overnight at 4 degrees. Reactions were quenched with 1 M Tris pH 7.4, and subsequently purified by SEC on HiLoad 16/600 Superdex 200 (GE Healthcare) or Superdex 200 10/300 GL (GE Healthcare) columns to remove residual biotin or fluorophore labels.

**Bead conjugation.** A 50% v/v slurry of 4.5–4.9 µm carboxyl-polystyrene particles (Spherotech) in PBS were mixed with FcεRIα-Ova fusion protein at ~2 mg/mL in a 1:1 v/v ration with EDC a 5 mg/mL (Thermo Scientific). The mixture was incubated at room with agitation for two hours, and the resultant conjugated beads were washed with 4 bead volumes of PBE buffer (PBS with 0.1% w/v BSA, 2.5 mM EDTA, with penicillin 100 U/mL and Streptomycin 100 µg/mL) and stored in PBE with sodium azide at a final concentration 0.02% w/v.

**Bead-based disruption assays.** FcεRIα-Ova coupled beads (0.25µl/assay-well of a 50% v/v bead suspension) were incubated for 2 h or overnight with 500 nM biotin-IgE-Fc$_{2–4}$, and washed four times with PBS to remove residual bIgE-Fc$_{2–4}$. Beads were resuspended to a final volume sufficient for 1 µl of slurry per assay-well. 20 µl of each anti-IgE agents was then serially diluted in a V-bottomed 96 well polystyrene plate at 105% the desired concentration, and 1µl of IgE+ beads were added to each well. Plates were incubated at room temperature with agitation (900 rpm) for 30 min prior to fixation with fresh 1.6% PFA in PBS. Fixed beads were then washed and stained for 15 min with 20 µl of a 1:400 dilution of PE/Cy5-Streptavidin (BioLegend, cat# 405205) in PBE prior to fixation with fresh 1.6% PFA in PBS. Plates were stored at 4 degrees prior to acquisition.

**Biolayer interferometry assays.** All BLI assays were conducted on an Octet-Red 96 (Fortebio). For anti-IgE SCK affinity measurements, biotin-labeled human IgE was immobilized on streptavidin (SA) tips to a density of ~0.5 nm. The tips were then exposed to a serial dilution of anti-IgE agents, starting at the lowest concentration, with association and dissociation times as indicated by experiment. Data was subtracted to blank reference tips exposed to the same serial analyte dilutions, aligned, and exported for analysis in BiaEval 3.0 (GE Healthcare) using a custom 1:1 SCK binding model without and RI-term as described in detail in Frenzel et al.[33]. Standard kinetic binding experiments with IgE fragments were also conducted on an Octet-Red 96 (Fortebio). For SCK binding studies with the locked-complex, the locked complex was capture coupled to Ni-NTA tips, coupled with EDC/NHS, quenched with 1 M ethanolamine, and residual nickel atoms were chelated from the resin with 500 mM EDTA pH8.0 prior to binding studies. The tips were then exposed to a serial dilution of anti-IgE agents, starting at the lowest concentration, with association and dissociation times as indicated by experiment. Data was subtracted to blank reference tips exposed to the same serial analyte dilutions, aligned, and exported for analysis in Clamp-XP using a custom 2:1 SCK binding model as described in detail in Karlsson et al.[34].

**BLI.** To test the ability of soluble omalizumab-IgG variants to disrupt IgE:FcεRIα complexes, FcεRIα-Ova was capture coupled to Ni-NTA tips, coupled with EDC/NHS, quenched with 1 M ethanolamine, and residual nickel atoms were chelated from the resin with 500 mM EDTA pH8.0. Tips were then saturated with recombinant human-IgE prior to incubation with 5µM anti-IgE agents for 15 min to measure dissociation.

**Yeast Culture.** Prior to transformation EBY100 yeast (ATCC) were grown in YPD media (10 g/L yeast extract, 20 g/L Peptone, 20 g/L Dextrose). Transformed yeast controls and libraries were propagated in SD-CAA media (6.7 g/L yeast nitrogen base with ammonium sulfate, 5.0 g/L casamino acids, 20 g/L Dextrose, buffered in pH 4.5 citrate buffer) and induced in SG-CAA (6.7 g/L yeast nitrogen base with ammonium sulfate, 5.0 g/L casamino acids, 2 g/L Dextrose, 18 g/L Galactose, buffered in pH 4.5 citrate buffer).

**Yeast surface staining and sorting.** Prior to selection yeast were induced for 24 h at 30 °C in SG-CAA. A tenfold representation of the estimated library diversity, or minimum of 100µl of overnight cultures, was stained prior to each round of selection assuming an approximate cell density of 1E7 cells per mL at OD600 of 1.0. During MACS selections yeast were subject to negative selection with magnetic anti-647 or streptavidin beads and LS columns (Miltenyi Biotec) and subsequently stained for selections as described in figures. After staining, yeast were washed in 50 mL PBE and resuspended in 5 mL PBE with magnetic binder beads (Miltenyi Biotec) for 15 min prior to positive selection over LS columns followed by one 5 mL PBE wash. Column bound yeast were eluted and enumerated on an Accuri Cytometer to estimate library recovery and resuspended at an OD600 < 1 in SD-CAA media for expansion. MACS selections were repeated until the estimated diversity was sufficiently low for FACS selections (1E5–1E6 estimated diversity). FACS selections were performed as indicated in figures on FACS Jazz Sorter (BD Biosciences) in the Stanford Shared Sorting Facility. For yeast efficiency stains yeast were exposed to precomplexed IgE:FcεRIα and AF-488 labeled G335C-IgE-Fc$_{3–4}$ for 30 min, washed, and stained with streptavidin AF-647 conjugate at a 1:400 dilution (ThermoFisher, MA, USA, cat# S21374) prior to acquisition or sorting. For comparative efficiency experiments in Fig. 3 between clones, yeast were stained following the same protocol, followed by 10 min of fixation in 1.6% PFA prior to secondary staining to control for differences in binding signals caused by acquisition and processing times. For all co-binding screens yeast were exposed to precomplexed IgE:FcεRIα at the indicated concentrations for 30 min, fixed in 1.6% PFA, and then labeled with streptavidin AF-647 conjugate at a 1:400 dilution (ThermoFisher) and FITC conjugated polyclonal rabbit anti-Ova antibodies at a 1:50 dilution (Rockland, PA, USA, cat# 200-4233-0100) followed by an additional

round of fixation. In all yeast experiments the cMyc tag was detected with poly-clonal chicken anti-cMyc antibodies at a 1:400 dilution (Gallus Immunotech, MA, USA cat# ACMYC) and PE conjugated polyclonal goat anti-chicken antibodies at a 1:200 dilution (EMD Millipore, MA, USA, cat# AP503H).

**Yeast library preparation**. Electrocompetent cells were produced using a modified lithium acetate (LiAc)/DTT protocol. A 300 mL culture of EBY100 yeast was grown to an OD of 1.6, treated for 15 min with 3 mL of 1 M DTT/1 M Tris and 15 mL of 2 M LiAc/TE solution, and then washed four times with ice cold electroporation buffer (1 M sorbitol, 1 mM CaCl₂). Insert DNA was amplified using pCTCon2_F and R primers (F: CGACGATTGAAGGTAGATA CCCATACGACGTTCCA-GACTACGCTCTGCAG R: CAGATCTCGAGCTATTACA AGTCCTCTTCA-GAAATAAGCTTTTGTTC) with the GeneMorph II random mutagenesis kit (Agilent), band purified, and further amplified with Platinum II Hot Start Mas-termix (ThermoFischer). 50µg of insert DNA was mixed with 10µg of triple-cut pCTCon2 vector (NheI, BamHI, SalI) and electroporated into electrocompetent yeast at 2500 V in 2 mm gap cuvettes (BioRad MicroPulser). Transformed cells were rescued in YPD for 1 h and then transferred to SD-CAA. Library diversity was estimated by number of transformants on serial diluted SD-CAA plates after 48 h.

**Protein crystallization**. All protein complexes were isolated by SEC, concentrated to 10 mg/ml and then screened using a Gryphon crystallization robot (Art Robbins Instruments). All hits from sitting drop screens were reproduced and optimized in hanging drops at 25 °C prior to harvesting and freezing. The final crystallization conditions for each complex are as follows: Omalizumab scFV: IgE-Fc₃₋₄ 0.02 M Magnesium chloride, 0.1 M HEPES pH 7.5, 22% (w/v) Polyacrylic acid 5100 sodium salt. 813:IgE-Fc₃₋₄: 0.1 M CAPS pH 10.5 40% (v/v) MPD. C02:IgE-Fc₃₋₄: 0.2 M Lithium sulfate, 0.1 M Phosphate citrate pH 4.2 20% (w/v) PEG 1000. HAE:IgE-Fc₃₋₄: 0.1 M Sodium acetate pH 4.6, 8% (w/v) PEG 4000. C02/HAE:IgE-Fc₃₋₄: 0.2 M Lithium sulfate, 0.1 M Sodium acetate pH 4.5, 50% (w/v) PEG 400.

**X-ray crystallography data collection and processing**. Data was collected at SLAC on beamlines 9-2 and 14-1, as well as at the Advanced Photon Source (APS) beamline 21-ID-F. Data was processed using X-Ray Detector Software (XDS)[35], solved by molecular replacement (Phenix), and subjected to rounds of model building and refinement (Phenix and Coot)[36–39].

**Cryo-EM data collection and processing**. 3.5 µL of purified complexes of clone_7 and locked-complex at a concentration of 4.5 mg/mL was applied to a glow-discharged holey carbon gold grid (Quantifoil R1.2/1.3, 200-mesh) with the addition of 0.01% octyl glucoside (OG). Excess protein was blotted away for 2 s with a force of 1 at room temperature in ~100% humidity before being plunge frozen into liquid ethane using a Vitrobot Mark IV (Thermo Fischer Scientific). Cryo-EM images were collected on a Titan Krios operated at 300 kV at a nominal magnification of 29,000× using a Gatan K3 direct electron camera in counting mode, corresponding to a pixel size of 0.8521 Å. Each image was composed of 40 frames with a dose rate of 16 electrons per pixel per second and an exposure time of 2 s, resulting in an accumulated dose of 44 electrons per Å². A total of 592 image stacks were acquired via the SerialEM 3–6 software suite[40] with a defocus range of −1.2 to −2.2 µm. Motion correction was performed with MotionCor2[41]. Initial particle sets were selected using crYOLO with a refined general model (gmodel_-phosnet_201912_N63.h5) and a total of 201,285 K particles were selected[42]. The remainder of data processing was conducted in cryoSPARC v3.0 using patch CTF and particle extraction[43]. Rounds of 2D classification, ab initio reconstruction, and heterogenous refinement were then used to sort and refine particle sets. The resultant models were used to generate templates for further template-based particle picking using cryoSPARC, and the resultant particles were subjected to rounds of 2D classification, ab initio reconstruction and heterogenous refinement to sort and refine classes. The best resultant class was homogenously refined, and subjected to non-uniform refinement.

**Cryo-EM model building**. Components of the disruption-intermediate complex were docked into the final map using Chimera[44]. The following existing models were used for docking: 1f6a (IgE Cε3, Cε4 and FcεRIα), 4j4p (Cε2), and C02:IgE-this study (scFv). The docked positions were then adjusted using dock-in-map (Phenix), and subjected to real-space refinement with reference and secondary structure restraints with default settings (Phenix)[36]. Carbohydrates were then automatically built into density maps when possible with Coot[38,39] and were not subject to further refinement.

**Bone marrow-derived mast cell (BMMC) cultures**. Culture, characterization and differentiation of transgenic genes hFcεRIα + BMMCs (BMMC^tg) from huFcεRIα-transgenic mice ((B6.Cg-Fcer1atm1Knt Tg(FCER1A)1Bhk/J) were derived by 3 to 4 weeks cultivation of bone marrow cells of the femur in BMMC culture medium (MC/9 media supplemented with 1 mM sodium pyruvate, 200mM L-glutamine, 1x non-essential amino acids, 50 µM β-Mercaptoethanol, 30 ng/ml recombinant IL-3 of the mouse). BMMCs were characterized by flow cytometry as CD117 + (clone

2B8, Thermo Fisher Scientific, MA, USA, 1:2000 dilution, cat# 12-1171-82), human FcεRIα + cells (antibody clone AER-37, Thermo Fisher Scientific, MA, USA, 1:1400 dilution, cat# 17-5899-42).

**Human basophil isolation**. Human peripheral whole blood was collected from volunteers with grass allergies who received informed consent in accordance with the Helsinki Declaration, and the study was approved by the Ethics Commission of the Canton of Bern, Switzerland and the Stanford University IRB. Primary human basophils were isolated from whole blood using Percoll density centrifugation of dextran-sedimented supernatants. Subsequently, the basophils were further enriched with negative selection using the Basophilic Isolation Kit II from Milteny (Miltenyi Biotec, Bergisch Gladbach, Germany). A human basophil culture medium (RPMI+/+) RPMI 1640 medium (Biochrome) was used, supplemented with 10% Hyclone FCS (Fisher Scientific), penicillin 100 U per mL, 100 µg per mL streptomycin and 10 mM HEPES buffer (Stock Solution 1 M, Life Technologies) and recombinant human IL-3 from Peprotec (United States).

**Disruptive potency and anaphylactogenicity assay with BMMCs^tg**. First, the BMMCs^tg were sensitized with 3 nM human NIP-specific IgE-JW8 (murine lambda chain) overnight at 37 °C in a 5% CO2 atmosphere. Following sensitization, the cells were washed three times with phosphate buffer saline (PBS) pH 7.4. Subsequently, disruptive anti-IgE inhibitors (156 nM–2500 nM), the positive activating anti-IgE Le27 (NBS-C Bioscience, Vienna, Austira), or an off-target humira control were diluted in supplemented BMMC culture medium MC/9 and added to the sensitized cells for 6 h at 37 °C in 5% CO2 atmosphere. The cells were then washed three times with PBS and divided into two fractions. The first fraction was used to measure the remaining surface IgE with a monoclonal anti-mouse anti-murine lambda chain - PE and the anaphylactogenicity by an activation marker staining. The second fraction of treated BMMCs^tg were stimulated with 100 ng/mL NIP(7)-BSA (Biosearch Technologies, INC) in the presence of an activation marker stain.

**Anaphylactogenicity, disruptive efficacy and inhibitory potency with human basophils**. To evaluate the anaphylactogenicity and the disruptive potency of the disruptive anti-IgE-DARPins we cultured isolated primary human basophils at a density of 1 × 10⁶ cells/ml per well in a round bottom plate with 96 wells in RPMI +/+ supplemented with 10 ng/ml recombinant human IL-3 (rhuIL-3) overnight at 37 °C. To assess the influence of IgE disruption on allergen induced basophil activation, we first defined the 6-grass allergen mix concentration (Bühlmann Laboratories AG, Schönenbuch, CH) at which suboptimal activation (ECSubopt.) can be achieved for each individual donor. For this purpose, we cultivated 25'000 basophils in RPMI+/+ medium, which was composed with the 6-grass allergen mix, IL-3, and the Bühlmann Flow CAST® Kit antibody stain (CD63-FITC and CCR3-PE), for 30 min at 37°. Subsequently, we measured activated cells (CD63+ cells) by flow cytometry. In the next step, 50,000 basophils were pre-incubated for 6 h with the disruptive anti-IgE inhibitors or Le27 and humira controls at 37 °C in RPMI+/+ medium with rhuIL-3. Upon incubation, the cells were washed three times with PBS and divided into two fractions. The first fraction was used to measure the remaining surface IgE with an anti-human IgE-PE antibody (clone Ige21, Thermo Fisher Scientific, MA, USA, 1:400 dilution, cat# 12-6986-42) and test for anaphylactogenicity with an anti-human CD63-FITC (Bühlmann Flow CAST® Kit antibody stain, 1:10 dilution, cat# FK-CCR). We used the second fraction to stimulate the cells with the predetermined ECSubopt of the 6-grass allergen mixture in the presence of a Flow CAST® kit antibody stain mix (activation staining of CD63). Finally, we determined activated basophil frequencies and surface measurements by flow cytometry.

**Figures and analysis**. Tables were generated using Microsoft Excel 14-6, and graphs and analysis were conducted using GraphPad Prism 8–9 unless otherwise noted.

**Reporting summary**. Further information on research design is available in the Nature Research Reporting Summary linked to this article.

## Data availability
There are no restrictions on data availability. All structures and models from this work have been deposited at the PDB (link: https://www.rcsb.org) under the following accession numbers: 7SHY, 7SI0, 7SHU, 7SHZ, 7SHT. Datasets from the PDB used in this work include 1F6A, 4GRG, 6NMS, 6NMU, 3B2U, 3C09, 3POY, 7MXI, 2Y7Q, 5G64, 1OOV, 5MOL, 2WQR, 5LGK, 4J4P, 6EYO. All main figures are derived from underlying raw data apart from representative flow cytometry gaiting plots. Source data are provided with this paper.

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

## Acknowledgements

We thank all members of the Eggel, Jardetzky, and Skiniotis laboratories involved in this study. We further acknowledge Dr. David Myszka for training in the analysis of alternate binding models in SCK experiments, and Pamela J. Focia for assistance with data collection for work conducted at APS. Use of the Stanford Synchrotron Radiation Light-source, SLAC National Accelerator Laboratory, is supported by the U.S. Department of Energy, Office of Science, Office of Basic Energy Sciences under Contract No. DE-AC02-76SF00515. The SSRL Structural Molecular Biology Program is supported by the DOE Office of Biological and Environmental Research, and by the National Institutes of Health, National Institute of General Medical Sciences (P41GM103393). We would also like to acknowledge E. Montabana, H. Hu, and D. Bushnell for microscope support; Y.-T. Lee and A. Seven for support with computing resources. Some of this work was performed at the Stanford-SLAC Cryo-EM Facilities, supported by Stanford University, SLAC and the National Institutes of Health S10 Instrumentation Programs. Data collection was also conducted at the Advanced Photon Source, a U.S. Department of Energy (DOE) Office of Science User Facility operated for the DOE Office of Science by Argonne National Laboratory under Contract No. DE-AC02-06CH11357. Cell sorting for this project was done on instruments in the Stanford Shared FACS Facility. We would also like to acknowledge Dr. Ana Rita Cardoso and Dr. Javaria Najeeb for supplying the control D10 antibody. Funding: This research was funded by a grant from the Fondation Acteria (to A.E.), the Research Fund of the Swiss Lung Association, Bern and the Uniscientia foundation (to A.E.), and NIH grants AI120510 (L.F.P. and T.S.J.) and AI115469 (to T.S.J.) and HL141493 (to T.S.J. and A.E.).

## Author contributions

L.F.P., T.S.J. and A.E. conceived the study. L.F.P., S.K., P.G., and C.Z. performed most experiments. All authors evaluated results and edited the manuscript. L.F.P., T.S.J and A.E. wrote the manuscript with input from all authors.

## Competing interests

L.F., A.E. and T.S.J. are shareholders and co-founders of Excellergy, a company developing new therapeutic molecules for multiple diseases. All other authors declare no competing interests.
