## [Peer Review File · Nature Communications]

Reviewers' Comments:

Reviewer #1:

Remarks to the Author:

The authors present a large body of experiments on the generation of omalizumab variants with increased displacing activity. This is combined with a structural analysis including crystal structures of complexes formed between IgE Fc and omalizumab variants and a cryoEM structure of a complex containing a fused receptor/IgE Fc protein and another omalizumab variant targeting the fusion protein.

Both parts of the paper are interesting work as such. The combination of both parts into one large manuscript, however, is somewhat artificial.

Major issues

Against the background of the low displacing activity of omalizumab, engineering of improved versions by evolutive techniques is a promising strategy. Here, the authors describe the development of omalizumab variants with increased displacing activity and efficacy. Authors have established a functional yeast display approach allowing selection of mutants for disruptive efficiency. As outcome of the extensive selection process they describe 2 variants, 813 "small increase", and C02 "five-fold" increase in disruptive activity.

Given that the authors claim to apply a disruptive efficiency stain, this increase does not seem impressive, in particular when considering that 813 is similar to parental omalizumab and the well established HAE variant, which is less disruptive than C02 but still more than 813. Another omalizumab variant described previously (i.a. by Davies, 2017, JBC)), also exhibits increased disruptive activity, but is not assessed comparatively in this manuscript and not really discussed at all in this manuscript.

Hence, omalizumab variants with enhanced disruptive activity have been described before, affecting to a certain degree the novelty.

In the later analysis of the variants as IgG using donor basophils and BMMCs, the authors claim that the C02_H2L2 variant significantly outperformed the HAE variant. In the light of the Figs 7e-h this statement is not justified. Further it has to be considered that the C02 variant before modification into the H2L2 variant definitely did not outperform the HAE variant (Fig 7a). Generally, the authors draw quite optimistic conclusions regarding the performance of their variants generated considering the data presented here.

The abstract claims that the presented work provide a framework for engineering disruptive inhibitors for other targets. It would then be appropriate to provide some examples of other protein based inhibitors in the discussion for which the described procedure may be adapted to. As it stands, the manuscript is too specific to claim general usability as experimental data is only presented for a single system.

The abstract describes cryo-EM structures of an IgE containing complex. In fact, the complex determined by cryo-EM contains the Fc fragment Ce2-4 of IgE, this is misleading. The (low) resolution of the cryo-EM reconstruction should also be stated in the abstract.

Multiple crystal structures are also presented of the complex between Omalizumab derivatives and Ig Fc Ce3-4. There are probably good experimental reasons for using these fragments rather than the intact IgE, but it would strengthen the manuscript if a proper introduction to as why IgE Fc Ce2-4 and IgE Fc Ce3-4 are valid proxies for the full IgE. Furthermore, a thorough account of the well established dynamic and conformational properties of IgE Fc must be provided.

Related to this, the number of references is unusually low for such a comprehensive and interdisciplinary set of experiments. As an example, three references are given in the materials and methods section. Hence, there is a significant risk that part of the experimental work cannot be reproduced outside the author research team. In addition, not a single reference is listed in the discussion, and the discussion is unusually short. This is very unfortunate, a much more elaborate

comparison with prior work is to be expected.

Examples of topics that needs to be discussed much more carefully:

What are the limitations of the disulfide bridged IgE Fc-FcεRIα, and what is the role of the Trp mutated to Cys in FcεRIα? Could this mutation contribute to the reorientation of the receptor observed by cryo-EM?

Can the conclusions based on the two IgE Fc constructs prepared for structural analysis be extrapolated to intact IgE without reservations?

Specific comments

The disruptive efficiency selection is quite complex. It seems to be a way of differentiating between Omalizumab scFv and the darpin as they behave the same in the 3 colors staining. This needs to be explained.

The authors relate the variants strongly to the darpins in Figs 2 and 3. This comparison should also be applied in later analysis, e.g. in Fig 7. How would a darpin and a bivalent darpin Fc variant perform in the cellular tests?

The disruptive efficiency KD/ID50 seems mainly driven by the ID50. In a previous paper from Eggel, the proteins could essentially be sorted in a very similar way just based on the ID50. On page 6 the authors claim further that KD and ID50 "could track linearly", however, all mutants have approximately the same K_a , so it seems driven by K_{diss} . On the same page the authors then hypothesise that these two parameters can be evolved independently. It remains unclear if the parameters are interdependent or not.

For the yeast display approach the authors use a concentration of 1.25 nM IgE Fc Ce3-4 closed conformation together with the complex. Why? Is that based on a titration?

The mutant IgEFc C3-4 used by the authors adopts a closed conformation (not mentioned) and is not equivalent to a free IgE Fc. So it might not reflect the affinity for free IgE Fc but more selectivity for this closed IgE Fc compared to the one from the complex.

The selection strategy aim to select mutants that bind more to the complex and less to the closed form => so even if they improve disruptive activity at the end, there is no selection on the disruption of the complex on the yeast.

P8. Reference to Fig 3f, is that perhaps Fig 3e-f?

P9. Biding -> Binding

P10 and Fig 4. The description of the intermolecular interfaces is too qualitative. Buried surface areas in the various structures should be presented. Panel d can be improved a lot, a higher magnification of the region containing D/E32, Y36 and R419 would be of interest. Key polar interactions should be indicated, at this resolution they should be reliable.

P10 "VL position 32" should be "VL position D/E 32"

P15 bottom. The conformational change of the Ce2 domain dimer should be quantitated with respect to other known structures containing the Ce2 dimer. Are there any signs of internal deviations from Ce2-Ce2 dimers present in other structures?

What is the estimated positional uncertainty in placement of the various domains in the cryo-EM 3D reconstruction?

The resolution of the cryo-EM is not impressive considering that a 300 kV scope with a K3 detector was used. Even though the complex has a molecular weight below 150 kDa, one could have

expected a higher resolution. The underlying reasons should be carefully discussed. Did the cryo-EM analysis provide indications of multiple conformations of the disruption intermediate?

Comments to the Figs and tables

In Figs 1,3, 5 the ID50, Kd, kd, ka are plotted in the style of laid out in Fig 1g. It would be simpler and give a more efficient presentation if small tables were used within each figure listing the values for these measured quantities. Also, such tables could include a column for disruptive efficiency, this would be helpful to the reader as the authors emphasize this quantity repeatedly.

Fig 1f : A control is needed that binds IgE but do not displace like Ligelizumab or E3_53 for expl and a negative ctrl not binding IgE. A table with ID50 values and statistic parameters (IC, Rmax ...) is needed.

Fig 1k : Efficiency of disruption was defined as Kd/ID50. Here it is the opposite. Same in text

Fig2a : Mutants improved on both KD and ID50, not only ID50, would be good too.

Fig2g-j : darpin and omalizumab seem to behave similarly, which should not be the case

Fig 3d: again omalizumab seems to be fully displacing. If it was on the 2D plot it probably would be a the same place as C02 and 813 (and Darpin).

Fig 5 c,d,e: The same symbols are used for different proteins, this is not optimal.

Fig 5f. This panel is not optimal and redundant with Fig 4a. Show a close-up of position 74 instead.

Fig 7 : Here a negative control without anti-IgE or with a non-relevant antibody like Humira. Moreover, it would be nice to see Darpin on this assay too.

FigS2 C-10: Does not correspond to Fig 3C, e.g. C02 is overlapping Darpin in Fig 3C but not in S2 C-10

Table S1. Rsym for omalizumab is 0.2864 (22.75). The last number is that a typo, and should it be 2.275?

It is stated at the bottom that data is from a single crystal. If that is true, the Rmerge is not relevant.

Material and methods.

In multiple instances, a space between number and unit is missing.

Are all proteins expressed in pCTcon2 for yeast display?

The authors state in Fig 2 that they incubated less than T1/2 but they never give incubation time for the selection or mention the expected T1/2

P28 Unit for MPD should be added, is it vol/vol? Likewise, the 8% PEG 4000 is missing a unit. X-ray crystallography, provide references for XDS and phenix.

P30 top. What specific steps in real space refinement were carried out? Were the carbohydrates included in refinement?

Reviewer #2:

Remarks to the Author:

General comments

The study by Pennington et al. is an interesting study that analyzes the mechanism of facilitated dissociation of IgE to its receptor and uses directed evolution of facilitated dissociation for the development of potential therapeutic molecules (anti-IgE antibodies) that can rapidly remove IgE and desensitize human effector cells. A yeast display approach was used to generate omalizumab variants that dissociate receptor-bound IgE. Different antibodies achieved enhanced potency via different paths (slower dissociation rates, larger association rates). These variants resulted in potentially therapeutic molecules that rapidly remove IgE and desensitize human allergy effector cells. The analysis of improved "disruptors" versus omalizumab are the main novelty and significance of the study and present a way to engineer disruptors for allergy therapy and other targets. The methods and results are well presented in detail and technically sound, although the use of some of the terminology can be improved as explained below.

Specific comments

Large versus small regions of overlap were attributed to inhibitors versus disruptors, respectively (Figure 1a-b). Since four structures were determined, it would be interesting to show the 4 surface sizes for the regions of overlap, as examples of how these differ in the three variant complexes versus the omalizumab complex.

The terminology used in describing the experiments should be improved to make it more consistent:

1-Page 7, first paragraph: lines 4 and 6 refer to "inhibitor", when it seems that they probably should refer to "disruptor" according to the terminology defined in figure 1a-b. The green coloring also is consistent with the coloring used in Figure 1b for the disruptor (anti-IgE). Maybe the term "disruptive inhibitors" in page 6, line 4th from the bottom, should be just "disruptors", to make the terminology consistent with Figure 1a-b.

2-Also, the terminology used in the second paragraph of page 7 is confusing: three anti-IgE were displayed in yeast. Those anti-IgE are disruptors and a non-competing DARPin. The use of "anti-IgE ligands" is confusing because in Figure 2b clearly the "ligand" is not what is displayed in yeast. Yeast displays "anti-IgE", not ligands. So, the term "anti-IgE molecules" would be better to describe what was displayed in yeast.

Figure 1: The molecules involved in the region of overlap needs to be explained, at least in the text or in the figure legend (is the overlap between the area on the receptor recognized by the inhibitor and the area recognized by the ligand?). In the bottom right image of Figure 1a, instead of "inhibitor blocks receptor", it probably should say -at least in the figure legend- "inhibitor blocks ligand binding to receptor". Explanation of section h needs to be completed ("Schematic of.") The colors for the structures in c-e need to be specified in the figure legend (e.g. magenta for receptor; blue for inhibitor and green for disruptor?).

Page 3, last paragraph: First sentence is ambiguous and should be completed. Omalizumab does not display disruptive activity at supraphysiologic (the meaning of this adjective should be explained) concentrations, but does it do it at other concentrations? Or never? Is it disruptive at certain concentrations?

Page 4, 4th line from the bottom: Does antigen mean "ligand" according to the Figure 1a-b? The molecule equivalent to ligand should be mentioned, for clarity purposes.

Page 16, last line: "HAE rests on the verge of cell activation" needs to be better explained, including the suggested mechanism for HAE that led to cell activation.

Page 18, line 15: the conclusion that the efficiency of disruption correlates with safety of anti-IgG disruptive agents needs to be further elaborated (safety in patients is not evaluated in this study, for example).

Movie 1 could not be found by the reviewer.

Minor comments

Page 5, line 16: we plotted

Page 20, last line and second line from bottom: "stained with" is repeated.

Page 9, 4th line from bottom: distal to the binding.

Meaning of green star in Fig 5f should be explained in legend.

Page 14, line 9: "each scFv and" ... VH?

Page 19, line 10: human allergy effector cells (cells are not "allergic").

Figure 2i legend should say: Schematic of co-binding screen? (instead of "stain"?).

Figure S2a-9. Meaning of "increase" and "decrease".

Fig S2 legend: b4: clone A03 (not A3).

Figure 4b: asterisks seem black, not magenta.

Reviewer #3:

Remarks to the Author:

General comments:

Pennington et al present a novel and in-depth exploration of antibody-mediated disruption of the complex between IgE and its receptor. The study has implications both for our fundamental understanding of protein-protein interactions and for antibody therapeutics of allergies.

A major strength of this work is the thorough application of in vitro selection and structural approaches to identifying and characterizing variants of omalizumab with an enhanced ability to bind the IgE:FceRI complex in a manner that facilitates complex dissociation. The use of a "locked" complex with an engineered disulfide bond between IgE and FceRI, which allowed the isolation and structural study of the "disruption intermediate" complex of an antibody, IgE, and FceRI, was particularly innovative and informative.

Specific comments:

- The authors define "disruptive efficiency", K_d/ID_{50} , and claim that it is "effectively a representation of the free energy barrier imposed on the anti-IgE inhibitor for binding to the receptor complex by the steric overlap with the receptor." The relation between disruptive efficiency and the free energy barrier needs further discussion and justification.
- Whereas the properties of antibodies studies and the three-dimensional structures of relevant complexes are documented in detail, the sequences of the antibody sequences need to be added to the Supplementary section.
- More detail is needed on the properties of the library created by error-prone PCR. In particular, what was the distribution of the location and the number of mutations relative to the parent clone?
- The authors state that "a combination of sequencing data and the efficiency screens isolated improved clones." The details of this workflow, including the sequences considered and the criteria used to choose the improved clones, need to be provided, at least in the Supplementary section.

We would like to thank the editor and reviewers for taking time to review our work. We have addressed the comments with additional experiments and data analysis and feel that the reviewers' input has helped us greatly improve the manuscript. Please find our point by point responses below in blue text in line with the original reviewer comments.

Reviewer #1, an expert in structural immunology (Remarks to the Author):

The authors present a large body of experiments on the generation of omalizumab variants with increased displacing activity. This is combined with a structural analysis including crystal structures of complexes formed between IgE Fc and omalizumab variants and a cryoEM structure of a complex containing a fused receptor/IgE Fc protein and another omalizumab variant targeting the fusion protein.

Both parts of the paper are interesting work as such. The combination of both parts into one large manuscript, however, is somewhat artificial.

While some of these studies are focused on screening and isolation of antibodies using multiple techniques, and others on structural studies, each section of the work informs the other and clarifies observations made in each section. Therefore we feel the data is best presented together. For example the observation that some stable intact complexes can bind inhibitors on yeast in screens, and the finding that two asymmetric epitopes are exposed in the cryo-EM structure, provides a unique mechanistic insight that one of the binding sites is unlikely to efficiently disrupt complexes. This observation related directly to the cellular safety studies. Hopefully with the clarified text in these revisions helps these relationships come across.

Major issues

Against the background of the low displacing activity of omalizumab, engineering of improved versions by evolutive techniques is a promising strategy. Here, the authors describe the development of omalizumab variants with increased displacing activity and efficacy. Authors have established a functional yeast display approach allowing selection of mutants for disruptive efficiency. As outcome of the extensive selection process they describe 2 variants, 813 "small increase", and C02 "five-fold" increase in disruptive activity.

Given that the authors claim to apply a disruptive efficiency stain, this increase does not seem impressive, in particular when considering that 813 is similar to parental omalizumab and the well established HAE variant, which is less disruptive than C02 but still more than 813. Another omalizumab variant described previously (i.a. by Davies, 2017, JBC)), also exhibits increased disruptive activity, but is not assessed comparatively in this manuscript and not really discussed at all in this manuscript.

Hence, omalizumab variants with enhanced disruptive activity have been described before, affecting to a certain degree the novelty.

In this work we wanted to modulate the efficiency and potency of disruption to help understand fundamental properties of disruptive antibodies. These studies, to our knowledge,

are the first studies that attempt to improve disruption directly, and the first to demonstrate that disruption and affinity can be engineered separately. While a 10-fold improvement in disruptive potency and a 5-fold improvement in efficiency may be less dramatic than changes observed in other parameters in other protein engineering manuscripts (e.g. affinity), we feel that the lack of improvement may be intrinsic to disruption.

In particular, in this specific case of omalizumab, we believe that limitations in disruptive efficiency likely arise because of structural constraints imposed by the omalizumab:IgE interaction. Without greatly changing the antibody binding footprint or binding pose on IgE, the predicted molecular overlap with the receptor remains constant and likely imposes a significant barrier to disruption. These engineering studies modulate how an antibody targets a trimolecular complex while still retaining the antibodies binding to the original epitope and we observe that increasing flexibility within the VH-VL packing plays a significant role in impact disruption. Importantly, even these relatively small changes have the potential to alter the function of these antibodies at relevant doses in human cells (Figure 7).

We would also like note that we were also able to dramatically reduce disruptive efficiency by engineering interfaces unique to the predicted trimolecular complex using the locked IgE:receptor complex (e.g. clone 7 in figure 5i). These observations show conclusively that disruptive efficiency and potency can vary dramatically across antibodies with similar affinities for IgE. These observations also demonstrate that disruptive antibodies have unique non-obvious features that can be engineered to modulate function.

The reviewer is correct to mention the excellent work by Davies et al. They described novel variants and scFv variants of omalizumab with ~2 fold improved affinity (1, 2). However little quantitative work was published about the ID50 of disruption or disruption efficiency in this work. We estimate that these variants exhibited a ~2-fold improved disruptive potency from single concentration SPR experiments, approximately in proportion to their improved affinity and therefore mirror our results with scFv of omalizumab closely. We have acknowledged this work now early in the results section when discussing scFv studies.

While we did not produce all variants from Davies et al. we did produce the external HAE variant previously engineered for high affinity IgE binding. We would like to highlight that prior to our current manuscript no description of HAE's disruptive activity or potency is present in the literature and it may not have been appreciated by scientists involved in its development. Our HAE observations therefore represent a novel contribution and highlight a previously unknown facet of HAE that we believe is critically important to understanding its adverse effects and failure in a phase 2 clinical trial.

In the later analysis of the variants as IgG using donor basophils and BMDCs, the authors claim that the C02_H2L2 variant significantly outperformed the HAE variant. In the light of the Figs 7e-h this statement is not justified.

While we did observe a statistically significant difference between HAE and C02 in human basophils (Fig 7i) we appreciate this point and modified the text to reflect the promising results of both C02 and HAE which have not been previously reported.

Original:

“Interestingly, C02_H2L2 significantly outperformed HAE in both basophil stripping and inhibition experiments and almost completely desensitized cells at a concentration of 500nM in 6 hours (Fig. 7i). In comparison, omalizumab had little effect on basophil IgE levels and signaling even at the highest concentrations studied.”

New:

“Interestingly, C02_H2L2 was consistently more potent than HAE in basophil stripping and inhibition experiments although both exhibited similar function in BMMCs^{tg}. Furthermore, both HAE and C02_H2L2 significantly desensitized cells at a concentration of 500nM in 6 hours (Fig. 7i). In comparison, omalizumab had little effect on basophil IgE levels and signaling even at the highest concentrations studied.”

Further it has to be considered that the C02 variant before modification into the H2L2 variant definitely did not outperform the HAE variant (Fig 7a). Generally, the authors draw quite optimistic conclusions regarding the performance of their variants generated considering the data presented here.

As pointed out by the reviewer we did elect to use the H2L2 variant of C02 as our most potent and safe antibody isolated in these studies. Unfortunately, the spontaneous activation observed in HAE_H2L2 variants prevented a similar analysis and comparison. We agree with the reviewer that the H2L2 modification enhances C02 function, and this observation is in line with our mechanistic studies that suggest conformational flexibility of antibodies is important for disruption. However we have adjusted our tone to highlight both antibodies in the text (See comment above).

The abstract claims that the presented work provide a framework for engineering disruptive inhibitors for other targets. It would then be appropriate to provide some examples of other protein based inhibitors in the discussion for which the described procedure may be adapted to. As it stands, the manuscript is too specific to claim general usability as experimental data is only presented for a single system.

We appreciate the reviewers point and removed the claim that the system is “generalizable” in the abstract, while keeping the statement that the system can “provide a conceptual framework for engineering new disruptive inhibitors for other targets.” We then expanded in the discussion to explain why we think the system should be generalizable and provide an example of a complex that could be targeted in this system by highlighting a known disruptor of CD47:SIRPa (also discussed in Figure 1d).

The abstract describes cryo-EM structures of an IgE containing complex. In fact, the complex determined by cryo-EM contains the Fc fragment Ce2-4 of IgE, this is misleading. The (low) resolution of the cryo-EM reconstruction should also be stated in the abstract.

We have adjusted the abstract as suggested by the reviewer.

Multiple crystal structures are also presented of the complex between Omalizumab derivatives and Ig Fc Ce3-4. There are probably good experimental reasons for using these fragments rather than the intact IgE, but it would strengthen the manuscript if a proper introduction to as why IgE Fc Ce2-4 and IgE Fc Ce3-4 are valid proxies for the full IgE. Furthermore, a thorough account of the well established dynamic and conformational properties of IgE Fc must be provided.

The fragments were used to facilitate crystallization and contain the full omalizumab epitope. There are virtually no differences in the omalizumab binding pose between the original omalizumab:IgE-Fc₃₋₄ fragment structure (3) and subsequent structures with the intact IgE-Fc₂₋₄ (1). We have added in the following text to reflect this:

“...in complex with a smaller IgE Fc fragment, IgE-Fc₃₋₄, which contains the full omalizumab epitope (Table S1).”

We also appreciate the reviewer’s comments about the dynamic nature of the IgE-Fc. We attempted to address one feature of this in an analysis of the open and closed dynamics of the IgE-Fc in relation to each scFv complex in the original manuscript (Fig. S4b and c). As the reviewer is aware the other dynamic portion of the IgE-Fc lies in the Ce2 domains, which are known to slow the dissociation of IgE from the high affinity receptor. However, we cannot evaluate the effect of each scFv variant on the positioning of the Ce2 domains in the partial IgE-Fc fragment structures lacking Ce2 domains. However, we agree this is an important dimension to address in our mechanistic studies and have expanded on our analysis of the Ce2 conformations in our section on the Cryo-EM structure as suggested by the review (See below).

We also modified the associated text to read:

“...Conformational rearrangements within the IgE-Fc have been extensively studied and are well known to effect receptor binding (4–7). Therefore we also assessed if scFv variants could modulate “open,” and “closed,” conformations of the IgE-Fc (4), as these conformation rearrangements are known to modulate IgE:FcεRIα interactions. This analysis revealed no correlation between the Fc conformational states and disruptive potency (Fig. S4b and c). Rearrangements of the Ce2 domains are also known to regulate receptor binding, however these domains are not present in these structures, and we cannot exclude the possibility that these scFvs differentially interact with or alter the orientation of these domains.”

Related to this, the number of references is unusually low for such a comprehensive and interdisciplinary set of experiments. As an example, three references are given in the materials

and methods section. Hence, there is a significant risk that part of the experimental work cannot be reproduced outside the author research team. In addition, not a single reference is listed in the discussion, and the discussion is unusually short. This is very unfortunate, a much more elaborate comparison with prior work is to be expected.

We have added additional citations to the discussion and text where appropriate, expanded the discussion to address several important issues raised by the reviewer (see below), and removed a citation in the methods addressing the production of a protein species (and added in a summary of the cited methods). The other 2 citations in the methods are in reference to kinetic models used to fit binding data and how to program these models into existing software packages. Here the papers provide extensive detailed methods, models, and in one case screen shots of the software interface to facilitate reproduction of the experiments- and we feel would add unnecessary length. We have also added in citations for programs used in methods where relevant.

Examples of topics that needs to be discussed much more carefully:

What are the limitations of the disulfide bridged IgE Fc-FcεRIa, and what is the role of the Trp mutated to Cys in FcεRIa? Could this mutation contribute to the reorientation of the receptor observed by cryo-EM?

We appreciate this point and we have addressed this question with the additional text below:

“Although these structural insights are consistent with our biochemical and biophysical studies, it is important to consider the possibility that the engineered disulfide bond used to capture the complex could alter the architecture of the complex itself. However, the spontaneous association of mutant IgE-Fc molecules with mutant FcεRIα receptors during co-expression, and the computational approach used to pick energetically favored disulfide bonds (25) suggest that the mutant complex is a reasonable surrogate for native complexes. Furthermore clone 7 not only bound locked complex tighter, but also stabilized native IgE:FcεRIα complexes in yeast based studies, suggesting that the locked complex reflects a transition state observed in native complexes.”

Can the conclusions based on the two IgE Fc constructs prepared for structural analysis be extrapolated to intact IgE without reservations?

The intact IgE-Fc used in cryo-EM studies has been previously shown to exhibit similar binding kinetics to the high affinity receptor as intact IgE (5). Likewise the IgE-Fc₃₋₄ fragment in complex with the omalizumab Fab was almost structurally identical to subsequent structures employing the intact IgE-Fc (3, 9). These molecules have served as experimental surrogates for receptor or IgE interactions in numerous publications and the conclusions of the paper have confined the interpretations of these structures to address questions can be addressed by the fragments alone. We did not feel that addressing this point will fit well into the discussion given the considerable length of the paper or change the reader’s interpretation of the data.

Specific comments

The disruptive efficiency selection is quite complex. It seems to be a way of differentiating between Omalizumab scFv and the darpin as they behave the same in the 3 colors staining. This needs to be explained.

We believe that our figures and labeling complicated this section. We have attempted to clarify this further to highlight the “efficiency” and “co-binding” stains used in this section. We have also modified the text to further clarify the method.

The authors relate the variants strongly to the darpins in Figs 2 and 3. This comparison should also be applied in later analysis, e.g. in Fig 7. How would a darpin and a bivalent darpin Fc variant perform in the cellular tests?

We appreciate this point; however we have previously published extensive studies on the function of E2_79 and other bivalent DARPins. While E2_79 and E3_53 were an important control in the protein engineering phase, the aim of this paper was to engineer and enhance omalizumab antibody function. We feel that the data presented clearly demonstrate these points and do not feel that a lack of comparison across all known anti-IgE agents detracts from these observations.

The disruptive efficiency $KD/ID50$ seems mainly driven by the $ID50$. In a previous paper from Eggel, the proteins could essentially be sorted in a very similar way just based on the $ID50$. On page 6 the authors claim further that KD and $ID50$ “could track linearly”, however, all mutants have approximately the same K_a , so it seems driven by K_{diss} . On the same page the authors then hypothesise that these two parameters can be evolved independently. It remains unclear if the parameters are interdependent or not.

In our manuscript we have attempted to understand how the disruptive potency of an antibody can be modulated and how this relates to various kinetic parameters in the case of omalizumab. Although each agent could be classified to an extent by the $ID50$ alone, in this manuscript we were interested in establishing routes to modify the disruptive potency in isolation to understand if changes in disruption required equivalent changes in binding kinetics. In this work we have shown that omalizumab variants with improved affinity appear to yield improved disruption proportionally. However by applying novel selection schemes (either disruption efficiency selection, or selections with the locked IgE receptor complex) we have shown that disruptive potency can change dramatically out of proportion to changes in binding affinity (or other kinetic parameters) to free IgE. We feel that clone C02 and Clone 7 both provide clear examples of scFv that exhibit changes in disruptive potency that do not track proportionately to changes in affinity. These results indicate that disruptive efficiency is a better parameter to characterize the nature of the transition state for disruption for a given inhibitor, effectively accounting for the difference in binding free energy of the inhibitor to the free and receptor-bound IgE.

As the reviewer highlights much of the change in affinity in our manuscript appears to be driven by changes in dissociation rate, however this analysis does not account for the activities of clone C02 and Clone 7. Our hypothetical examples and comparisons to experimental data are intended to highlight some (but not all) of these relationships and have focused on affinity for simplicity. However, in response to the reviewer's comment, in each section we made changes in the text to note the slower dissociation rates and we have provided plots throughout the manuscript to highlight relationships of all kinetic parameters for the reader to review.

For the yeast display approach the authors use a concentration of 1.25 nM IgE Fc Ce3-4 closed conformation together with the complex. Why? Is that based on a titration?

The reviewer makes an important point here. These values were based on a titration as suggested and we now note in the text that we used a titration to establish non-saturating concentrations of the inactive-ligand in the disruptive efficiency screen .

At very high concentrations of the inactive IgE species (G335C-IgE-Fc₃₋₄) E2_79 and omalizumab yeast are saturated with the inactive IgE species and disrupt/bind little of the biotin labeled IgE-Fc (bIgE-Fc₂₋₄) saturated with FceRIa. In contrast yeast displaying the non-competitive binder E3_53 bind robustly to FceRIa saturated IgE in the presence of higher concentrations of G335C-IgE-Fc₃₋₄ as E3_53 can target free and receptor bound IgE with similar affinities.

The mutant IgEFc C3-4 used by the authors adopts a closed conformation (not mentioned) and is not equivalent to a free IgE Fc. So it might not reflect the affinity for free IgE Fc but more

selectivity for this closed IgE Fc compared to the one from the complex.

The reviewer makes an important point. We have previously published comparative binding studies and shown that omalizumab and E2_79 exhibit similar affinities for the closed (G335C-IgE-Fc₃₋₄) and open (IgE-Fc₃₋₄) IgE Fc fragments and shown that E2_79 and omalizumab have an epitope confined to the stable Ce3 domain of IgE (10, 11). We therefore feel the closed IgE-Fc is a fair surrogate for free IgE. However, we cannot rule out the possibility that we select for variants that target open IgE-Fc conformations, and this conformational bias could be tied to their improved disruptive efficiency. Therefore we have added the following text:

“We have previously shown that omalizumab and E2_79 bind with similar affinity to WT and G335C mutant IgE-Fc₃₋₄ fragments and have an epitope confined to the stable Cε3 domain (3, 8). Nevertheless it is important to consider that the conformational state of the inactive ligand mutant could favor the selection of antibodies with a conformational bias.”

The selection strategy aim to select mutants that bind more to the complex and less to the closed form => so even if they improve disruptive activity at the end, there is no selection on the disruption of the complex on the yeast.

We understand why the reviewer is making this point and we acknowledge that we could also select for variants that only “bind more to complex.” However in our efficiency assay binding more complex can occur either by disrupting complexes and trapping IgE or by binding to intact IgE:receptor complexes. In our second co-binding assay we check for the presence of FcεRIa to confirm if the variants bound intact complexes or disrupted and trapped IgE. Both hits which act as robust “co-binders” and strong “disruptors” can be appreciated in this second assay (for example C02 vs Clone 7 in figure 5h). These categorizations match the data produced with soluble scFvs and our efficiency selections clearly isolated more disruptive clones. Therefore we think there is strong experimental evidence that we do directly select and screen for disruption. This is not to say that we do not simultaneously select for clones that target more open IgE-Fc conformations (as the IgE in the receptor complex is open and our G335C and WT free IgE exist in a closed state) however it is possible that these selections go hand in hand. However we would like to highlight that crystal structures with variants showed no bias towards a specific IgE-Fc conformational state (Figure S4b).

P8. Reference to Fig 3f, is that perhaps Fig 3e-f?

We have revised this section to clarify these references.

P9. Biding -> Binding

We have changed this.

P10 and Fig 4. The description of the intermolecular interfaces is too qualitative. Buried surface

areas in the various structures should be presented. Panel d can be improved a lot, a higher magnification of the region containing D/E32, Y36 and R419 would be of interest. Key polar interactions should be indicated, at this resolution they should be reliable.

As requested, we have added in the buried surface area analysis (Figure 4c) and a detailed view showing the hydrogen bonding network around R419 (Figure S4d). However we feel that panel d serves an important purpose as is to demonstrate that the overall architecture of the antibody changed very little during affinity maturation. While figure 4 represents a tremendous amount of experimental effort and potential data we have intentionally limited the size of the figure as it mostly presents negative data. The structures are primarily useful to demonstrate that changes in disruption cannot be easily inferred from the interactions of that antibody and ligand alone out of the context of the receptor.

We have also expanded figure 4 to highlight how the unperturbed binding pose of the antibodies did not change the overlap of the antibodies and the receptor, a critical parameter we had previously shown to relate to disruptive efficiency (Figure 4b), at the request of another reviewer.

P10 “VL position 32” should be “VL position D/E 3

Changed

P15 bottom. The conformational change of the Ce2 domain dimer should be quantitated with respect to other known structures containing the Ce2 dimer. Are there any signs of internal deviations from Ce2-Ce2 dimers present in other structures?

As the reviewer suggests the Ce2 domain is conformationally dynamic across structures. We attempted to further clarify the text associated with this comment to highlight that the position of the Ce2 dimer observed in our cryo-EM structure is novel and distinct from that observed in omalizumab:IgE-Fc complexes and other structures (as shown in figure 6g).

Per the reviewer’s request we have also pulled all published structures with Ce2 domain present, aligned these by Ce3 domains proximal to the Ce2 dimer, and quantified the relative displacement of Ce2 dimers relative to the first published structure of the intact IgE-Fc. This can be found in supplementary Table S9.

What is the estimated positional uncertainty in placement of the various domains in the cryo-EM 3D reconstruction?

We have added in the following graphs to illustrate the correlation coefficient (CC) of each chain by residue to highlight areas with poorer fit (e.g. low resolution Ce2 domain in IgE chains or the IgE distal regions of FcεRIα).

The resolution of the cryo-EM is not impressive considering that a 300 kV scope with a K3 detector was used. Even though the complex has a molecular weight below 150 kDa, one could have expected a higher resolution. The underlying reasons should be carefully discussed. Did the cryo-EM analysis provide indications of multiple conformations of the disruption intermediate?

We believe that the resolution was limited by time, access, and the conformational heterogeneity of the complex. During processing the Cε2 domains were not apparent until after multiple rounds of heterogeneous refinement identified a subset of particles that refined to a higher resolution (Cryosparc equivalent to 3D classification). Furthermore both the scFv and high affinity receptor are significantly displaced from their native binding positions and remain only partially bound. Therefore it is not clear if expanding the size of this small dataset (~500 images) would yield significant gains in resolution. Attempts to quantify heterogeneous conformational states in the data using 3D variability analysis did not identify major distinct states-likely due to the small size of each mobile domain. Non-uniform and local refinement also failed to dramatically improve the map quality despite attempting a range of masks for mobile portions of the model. We have added a brief section discussing this in the discussion.

Comments to the Figs and tables

In Figs 1,3, 5 the ID50, Kd, kd, ka are plotted in the style of laid out in Fig 1g. It would be simpler and give a more efficient presentation if small tables were used within each figure listing the values for these measured quantities. Also, such tables could include a column for disruptive efficiency, this would be helpful to the reader as the authors emphasize this quantity repeatedly.

In an effort to reduce tables within figures, and to provide a clear graphical relationship of trends in the potency and affinity changes (and thus relative efficiencies) we have elected to plot the data in the figures. We feel strongly that this enhances the clarity of the story by linking

the data to schematic diagrams and allowing the reader to appreciate several complex relationships. However we also agree with the reviewers point that a tabular representation of the data would be useful and simple and have elected to place all affinity and potency tables as supplementary tables adjacent to one another for reference.

Fig 1f : A control is needed that binds IgE but do not displace like Ligelizumab or E3_53 for expl and a negative ctrl not binding IgE. A table with ID50 values and statistic parameters (IC, Rmax ...) is needed.

We have added these controls (now figure S1D) and provides tables with numerical values in the supplemental files.

Fig 1k : Efficiency of disruption was defined as K_d/ID_{50} . Here it is the opposite. Same in text

We have fixed this discrepancy.

Fig2a : Mutants improved on both K_D and ID_{50} , not only ID_{50} , would be good too.

I think we agree on this point and show this idea with several omalizumab clones that have proportional increases in the affinity and ID_{50} (and thus have a similar efficiency). We have attempted to clarify the text around this section as well. This would be the case $A \rightarrow B$ where the efficiency stays the same while potency and affinity to free ligand increase.

Fig2g-j : darpin and omalizumab seem to behave similarly, which should not be the case

These figures highlight several different assays including: 1. a pure disruption screen, 2. our efficiency screen, and 3. Our co-binding screen. In figure 2 g and h(left panel) the control reagents are exposed to intact complexes alone. This assay most closely reflects disruptive potency and can differentiate E2_79 and omalizumab to an extent (although the magnitude of the difference is small compared to the very large binding signal of E3_53 in the accompanying figures). However this system is attempting to recreate a phenomenon observed with soluble anti-IgE molecules at high concentrations with tethered anti-IgE molecules on the surface of yeast. Therefore it is not surprising that the system displays differences to soluble protein disruption assays. Notably the efficiency stain 2h (right panel) clearly shows how the addition of the G335C-IgE improves the discrimination of these controls. This reagent was primarily used to a control for free IgE affinity during selections to drive disruptive efficiency and in these experiments show how this additional dimension improves the segregation of E2_79 and omalizumab which each have distinct disruptive efficiencies.

Finally in figure 2J we are evaluating how much FcεR1a remains bound to IgE species captured by yeast. In this assay we are trying to distinguish disruptive agents from non-competitive agents and we would expect E2_79 and omalizumab to appear similar as each should primarily disrupt complexes and promote the dissociation of FcεR1a.

Nevertheless we feel that the reviewers' comments here and elsewhere highlight deficiencies in the text and figures in describing selections and we have attempted to clarify these points with additional labels and revised text.

Fig 3d: again omalizumab seems to be fully displacing. If it was on the 2D plot it probably would be at the same place as C02 and 813 (and Darpin).

Per the comment above this figure is not showing the efficiency selection but is showing the co-binding screen. We have added labels to each assay to clear up confusion. The 2D plots of the efficiency selections are presented in the preceding panel and show the difference across omalizumab variants. We would expect most disruptive agents displayed on yeast to retain little receptor bound to the IgE species they capture. Nevertheless we do see that higher affinity potent disrupters can bind intact complexes to a degree (albeit much less than non-competitive controls like E3_53). We have added text and a panel to figure 2 to address the important observation. Furthermore this phenomenon is addressed in subsequent sections and is likely a result of the two asymmetric binding sites identified in the cryo EM structure, where one C2 proximal site may not rapidly induce disruption but allow yeast to bind intact complexes.

Fig 5 c,d,e: The same symbols are used for different proteins, this is not optimal.

We have adjusted this accordingly.

Fig 5f. This panel is not optimal and redundant with Fig 4a. Show a close-up of position 74 instead.

We have adjusted this accordingly to show residue 74.

Fig 7 : Here a negative control without anti-IgE or with a non-relevant antibody like Humira. Moreover, it would be nice to see Darpin on this assay too.

In figure 7a we have now included an off target anti-CoV2 antibody (D10) and the non-competing DARPin E3_53 control in the supplemental figure S1 of the paper relating to the introduction to the bead based assay. For figure 7d we have repeated BLI experiments with the corresponding control D10-IgG and DARPin E3_53 controls and included these in Figure S5I. Finally we have also included the Humira negative control and crosslinking anti-IgE Le27 positive control for both BMMC and whole human blood anaphylactogenicity experiments in figure 7b and c.

We have published extensively characterizing the DARPins in human cells and BMMCs. While a head-to-head comparison would have been interesting, we feel it is not the primary aim of the paper which focuses on the evolution of omalizumab.

FigS2 C-10: Does not correspond to Fig 3C, e.g. C02 is overlapping Darpin in Fig 3C but not in S2

We appreciate the reviewers point and realized that we have left a key piece of information out of the figures and methods. In figure 3C-G, for the purpose of providing a more quantitative comparison, the individual sequence confirmed plasmid controls were transformed to fresh EBY100 cultures, colonies were selected, and cultures were induced on the same day. Furthermore, to ensure that time to acquire samples did not affect staining intensities for the more quantitative comparisons in figure 3, samples were fixed after staining. As such the staining intensity in these samples is improved for all controls and variant hits. This is an even more critical experimental detail for the co-binding screen, as the signal observed for the disruptive variants is transient and requires fixation for clear visualization. This is why the co-binding assays were conducted as a screen and not directly during selections where fixation would have killed the cells (only a true non-competitive inhibitor such as E3_53 gave a robust and stable signal without fixation).

We have now noted this issue in the written methods, text, and in the figure legend here and figure S2. We have provided the raw data for Fig. S2C below to highlight how the plot reflects the flow data. We believe that the slightly slower dissociation of hits like C02, and the higher expression surface density and subsequent bivalent binding events, reduced the loss in unfixed samples during the ~15 minutes of secondary staining and 1 hour acquisition per plate and thus the controls may have lost IgE signal more rapidly. We would like to point out that the controls in the context of selections are primarily used to confirm that gross pattern of staining, and the selections themselves are conducted using gates on the library itself- therefore these relative differences appeared to have little effect on selections.

Sample Name	Subset Name	Count
C02.fcs	C335	2647
E07 E3_53nxT.fcs	C335	3293
B07 Oma.fcs	C335	2895
A07 E2_79.fcs	C335	2529

Table S1. Rsym for omalizumab is 0.2864 (22.75). The last number is that a typo, and should it be 2.275?

We have fixed this error.

It is stated at the bottom that data is from a single crystal. If that is true, the Rmerge is not relevant.

We have fixed this error carried over from the table template.

Material and methods.

In multiple instances, a space between number and unit is missing.

We have fixed all units to include the appropriate space.

Are all proteins expressed in pCTcon2 for yeast display?

Yes all proteins have been displayed in the pCTCon2 vector.

The authors state in Fig 2 that they incubated less than T1/2 but they never give incubation time for the selection or mention the expected T1/2

The incubation time is 30 minutes. We have revised the methods to include this detail.

P28 Unit for MPD should be added, is it vol/vol? Likewise, the 8% PEG 4000 is missing a unit. X-ray crystallography, provide references for XDS and phenix.

We have added the units and appropriate citations.

P30 top. What specific steps in real space refinement were carried out? Were the carbohydrates included in refinement?

We have updated the methods to include these details

Reviewer #2, an expert in IgE biology (Remarks to the Author):

General comments

The study by Pennington et al. is an interesting study that analyzes the mechanism of facilitated dissociation of IgE to its receptor and uses directed evolution of facilitated dissociation for the development of potential therapeutic molecules (anti-IgE antibodies) that can rapidly remove IgE and desensitize human effector cells. A yeast display approach was used to generate omalizumab variants that dissociate receptor-bound IgE. Different antibodies achieved enhanced potency via different paths (slower dissociation rates, larger association rates). These variants resulted in potentially therapeutic molecules that rapidly remove IgE and desensitize human allergy effector cells. The analysis of improved “disruptors” versus omalizumab are the main novelty and significance of the study and present a way to engineer disruptors for allergy

therapy and other targets. The methods and results are well presented in detail and technically sound, although the use of some of the terminology can be improved as explained below.

Specific comments

Large versus small regions of overlap were attributed to inhibitors versus disruptors, respectively (Figure 1a-b). Since four structures were determined, it would be interesting to show the 4 surface sizes for the regions of overlap, as examples of how these differ in the three variant complexes versus the omalizumab complex.

We agree that this analysis could provide an interesting insight into mechanism of dissociation across the structures. As outlined in Figure 1a-b variation in the volume of overlap does seem to be the most critical parameter, and most reflective of steric clashes between molecules. Per the reviewers request we have included the analysis of the overlapping volumes in a revised version figure 4 to address this point. As suggested by the original text, but not shown, these overlaps are extremely similar across variants and do not appear to explain the differences in disruption.

The terminology used in describing the experiments should be improved to make it more consistent:

1-Page 7, first paragraph: lines 4 and 6 refer to “inhibitor”, when it seems that they probably should refer to “disruptor” according to the terminology defined in figure 1a-b. The green coloring also is consistent with the coloring used in Figure 1b for the disruptor (anti-IgE). Maybe the term “disruptive inhibitors” in page 6, line 4th from the bottom, should be just “disruptors”, to make the terminology consistent with Figure 1a-b.

This is a good point and we have attempted to rework the diagrams for clarity. In some instances we are discussing a displayed binder that might be either a disruptive or competitive inhibitor. Therefore we have adjusted the figure to refer to disruptive or competitive inhibitors and used inhibitors alone in the text where we are discussing general cases.

2-Also, the terminology used in the second paragraph of page 7 is confusing: three anti-IgE were displayed in yeast. Those anti-IgE are disruptors and a non-competing DARPin. The use of “anti-IgE ligands” is confusing because in Figure 2b clearly the “ligand” is not what is displayed in yeast. Yeast displays “anti-IgE”, not ligands. So, the term “anti-IgE molecules” would be better to describe what was displayed in yeast.

We agree and have changed the text for clarity.

Figure 1: The molecules involved in the region of overlap needs to be explained, at least in the text or in the figure legend (is the overlap between the area on the receptor recognized by the inhibitor and the area recognized by the ligand?).

We have added a section to the figure legend to explain that this is a volume overlap corresponding to overlapping region between two maps generated from the superimposed models. For example the SIRP α :CD47 complex was aligned to the structure of a competitive (6NMS) or disruptive (6NMU) antibody SIRP α complex. A map was then generated for CD47 and each antibody and the relative overlap of the antibody and CD47 maps is shown as a orange volume. In this way the orange region reflects the estimated volume overlap between the antibody and receptor.

In the bottom right image of Figure 1a, instead of “inhibitor blocks receptor”, it probably should say -at least in the figure legend- “inhibitor blocks ligand binding to receptor”.

We agree and have changed the figure for clarity.

Explanation of section h needs to be completed (“Schematic of.”)

Thank for bringing this error to our attention. We have filled this out.

The colors for the structures in c-e need to be specified in the figure legend (e.g. magenta for receptor; blue for inhibitor and green for disruptor?).

This has been added to the corresponding figure legend.

Page 3, last paragraph: First sentence is ambiguous and should be completed. Ligelizumab does not display disruptive activity at supraphysiologic (the meaning of this adjective should be explained) concentrations, but does it do it at other concentrations? Or never? Is it disruptive at certain concentrations?

We have clarified this point and changed the sentence to read:

“Omalizumab displays weak disruptive activity at high concentrations, while ligelizumab does not display any disruptive activity (11, 12), despite the ~100-fold higher affinity of ligelizumab for IgE.”

Page 4, 4th line from the bottom: Does antigen mean “ligand” according to the Figure 1a-b? The molecule equivalent to ligand should be mentioned, for clarity purposes.

We have changed this for clarity to read:

“This structure represents a “transition state” intermediate along the disruption pathway and reveals unanticipated interactions outside of the known antibody:antigen interface that modulate both affinity and the rate of disruption.”

Page 16, last line: “HAE rests on the verge of cell activation” needs to be better explained, including the suggested mechanism for HAE that led to cell activation.

We have added the following text to address this point:

“Given the similar disruption profiles of HAE_IgG to the HAE_H2L2_IgG in bead based disruption experiments, this data suggests that even small changes in the balance of binding and disruption events at the cell surface may favor receptor crosslinking and promote cell activation. These data may indicate that HAE rests on the verge of cell activation, consistent with sporadic hypersensitivity events that terminated phase 2 clinical trials (13) and warrants further investigation in more subjects.”

Page 18, line 15: the conclusion that the efficiency of disruption correlates with safety of anti-IgG disruptive agents needs to be further elaborated (safety in patients is not evaluated in this study, for example).

We have added the following text:

“These results suggest that efficient disruptors that bind with lower affinity relative to their ID50 rapidly disrupt complexes or rapidly bind and dissociate from complexes without disruption. In contrast higher affinity less efficient disruptors bind longer prior to disruption or dissociation. In the case of IgE:FcεR1α complexes these longer dwell times increase the probability of antibody induced crosslinking of receptors and unwanted spontaneous activation of basophils.”

Movie 1 could not be found by the reviewer.

We have reattached this file.

Minor comments

Page 5, line 16: we plotted **Fixed**

Page 20, last line and second line from bottom: “stained with” is repeated. **Fixed**

Page 9, 4th line from bottom: distal to the binding. **Fixed**

Meaning of green star in Fig 5f should be explained in legend **Per another reviewer request we modified this figure to include a clear detail of the indicated position)**

Page 14, line 9: “each scFv and” ... VH? **Fixed**

Page 19, line 10: human allergy effector cells (cells are not “allergic”). **We appreciated this grammatical point, however the term allergic effector cells is used throughout literature to refer to cells involved in allergy. The adjective allergic mean caused by or related to an allergy so we feel it is appropriate to use in this context as well.**

Figure 2i legend should say: Schematic of co-binding screen? (instead of “stain”?). **Fixed**

Figure S2a-9. Meaning of “increase” and “decrease”. **Fixed to indicate frequency**

Fig S2 legend: b4: clone A03 (not A3). **Fixed**

Figure 4b: asterisks seem black, not magenta. **Fixed**

Reviewer #3, and expert in protein engineering and display technologies (Remarks to the Author):

General comments:

Pennington et al present a novel and in-depth exploration of antibody-mediated disruption of the complex between IgE and its receptor. The study has implications both for our fundamental understanding of protein-protein interactions and for antibody therapeutics of allergies.

A major strength of this work is the thorough application of in vitro selection and structural approaches to identifying and characterizing variants of omalizumab with an enhanced ability to bind the IgE:FceRI complex in a manner that facilitates complex dissociation. The use of a “locked” complex with an engineered disulfide bond between IgE and FceRI, which allowed the isolation and structural study of the “disruption intermediate” complex of an antibody, IgE, and FceRI, was particularly innovative and informative.

Specific comments:

- The authors define “disruptive efficiency”, $K_d/ID50$, and claim that it is “effectively a representation of the free energy barrier imposed on the anti-IgE inhibitor for binding to the receptor complex by the steric overlap with the receptor.” The relation between disruptive efficiency and the free energy barrier needs further discussion and justification.

We have added a new figure (now supplemental figure 1a) to illustrate this concept. This figure illustrates how the free energy barrier of disruption is lower than that of spontaneous complex dissociation, a key concept which underlies the accelerated dissociation. The figure also highlights how the free energy barrier to inhibitor binding to complexes is effectively the energy barrier to disruption. This barrier is likely proportional to the extent of steric overlap, and as we demonstrate here with clone 7, it is also dependent upon transient interactions that can form during the transition state. More efficient disruptors would be predicted to have lower energy barriers to binding. In this framework the $\Delta\Delta G$ of disruption would relate to the magnitude of improvement in disruptive efficiency.

- Whereas the properties of antibodies studies and the three-dimensional structures of relevant complexes are documented in detail, the sequences of the antibody sequences need to be added to the Supplementary section.

Per the Journals request the antibody sequences have been deposited online with their corresponding structures. All antibodies not featured in deposited structures have minimal amino acid substitutions and these have been indicated in the relevant tables.

- More detail is needed on the properties of the library created by error-prone PCR. In

particular, what was the distribution of the location and the number of mutations relative to the parent clone?

We did not conduct deep sequencing of the libraries, and given that error prone PCR was employed we can only estimate the frequency of mutations predicted by the protocols associated with the EP polymerase. The mutations should have been randomly distributed throughout the amplified scFv construct.

- The authors state that “a combination of sequencing data and the efficiency screens isolated improved clones.” The details of this workflow, including the sequences considered and the criteria used to choose the improved clones, need to be provided, at least in the Supplementary section.

We have provided a very extensive step by step outline of this process in supplementary figure 2 and attempted to clarify the figure as needed in the revised manuscript.

1. A. M. Davies, E. G. Allan, A. H. Keeble, J. Delgado, B. P. Cossins, A. N. Mitropoulou, M. O. Y. Pang, T. Ceska, A. J. Beavil, G. Craggs, M. Westwood, A. J. Henry, J. M. McDonnell, B. J. Sutton, Allosteric mechanism of action of the therapeutic anti-IgE antibody omalizumab, *J. Biol. Chem.* (2017), doi:10.1074/jbc.M117.776476.
2. A. N. Mitropoulou, T. Ceska, J. T. Heads, A. J. Beavil, A. J. Henry, J. M. McDonnell, B. J. Sutton, A. M. Davies, Engineering the Fab fragment of the anti-IgE omalizumab to prevent Fab crystallization and permit IgE-Fc complex crystallization, *Acta Crystallogr. Sect. F Struct. Biol. Commun.* (2020), doi:10.1107/S2053230X20001466.
3. L. F. Pennington, S. Tarchevskaya, D. Brigger, K. Sathiyamoorthy, M. T. Graham, K. C. Nadeau, A. Eggel, T. S. Jardetzky, Structural basis of omalizumab therapy and omalizumab-mediated IgE exchange, *Nat. Commun.* **7** (2016), doi:10.1038/ncomms11610.
4. B. A. Wurzburg, T. S. Jardetzky, Conformational flexibility in immunoglobulin E-Fc 3-4 revealed in multiple crystal forms., *J. Mol. Biol.* **393**, 176–90 (2009).
5. M. D. Holdom, A. M. Davies, J. E. Nettleship, S. C. Bagby, B. Dhaliwal, E. Girardi, J. Hunt, H. J. Gould, A. J. Beavil, J. M. McDonnell, R. J. Owens, B. J. Sutton, Conformational changes in IgE contribute to its uniquely slow dissociation rate from receptor FcεRI., *Nat. Struct. Mol. Biol.* **18**, 571–6 (2011).
6. F. Jabs, M. Plum, N. S. Laursen, R. K. Jensen, B. Mølgaard, M. Miehe, M. Mandolesi, M. M. Rauber, W. Pfützner, T. Jakob, C. Möbs, G. R. Andersen, E. Spillner, Trapping IgE in a closed conformation by mimicking CD23 binding prevents and disrupts FcεRI interaction, *Nat. Commun.* (2018), doi:10.1038/s41467-017-02312-7.
7. B. Dhaliwal, D. Yuan, M. O. Y. Pang, A. J. Henry, K. Cain, A. Oxbrow, S. M. Fabiane, A. J. Beavil, J. M. McDonnell, H. J. Gould, B. J. Sutton, Crystal structure of IgE bound to its B-cell receptor CD23 reveals a mechanism of reciprocal allosteric inhibition with high affinity receptor FcεRI., *Proc. Natl. Acad. Sci. U. S. A.* **109**, 12686–91 (2012).
8. D. B. Craig, A. A. Dombkowski, Disulfide by Design 2.0: A web-based tool for disulfide

- engineering in proteins, *BMC Bioinformatics* (2013), doi:10.1186/1471-2105-14-346.
9. A. M. Davies, E. G. Allan, A. H. Keeble, J. Delgado, B. P. Cossins, A. N. Mitropoulou, M. O. Y. Pang, T. Ceska, A. J. Beavil, G. Craggs, M. Westwood, A. J. Henry, J. M. McDonnell, B. J. Sutton, Allosteric mechanism of action of the therapeutic anti-IgE antibody omalizumab, *J. Biol. Chem.* (2017), doi:10.1074/jbc.M117.776476.
10. P. Gasser, S. S. Tarchevskaya, P. Guntern, D. Brigger, R. Ruppli, N. Zbären, S. Kleinboelting, C. Heusser, T. S. Jardetzky, A. Eggel, The mechanistic and functional profile of the therapeutic anti-IgE antibody ligelizumab differs from omalizumab, *Nat. Commun.* (2020), doi:10.1038/s41467-019-13815-w.
11. B. Kim, A. Eggel, S. S. Tarchevskaya, M. Vogel, H. Prinz, T. S. Jardetzky, Accelerated disassembly of IgE-receptor complexes by a disruptive macromolecular inhibitor, *Nature* (2012), doi:10.1038/nature11546.
12. A. Eggel, G. Baravalle, G. Hobi, B. Kim, P. Buschor, P. Furrer, J.-S. Shin, M. Vogel, B. M. Stadler, C. A. Dahinden, T. S. Jardetzky, Accelerated dissociation of IgE-FcεRI complexes by disruptive inhibitors actively desensitizes allergic effector cells., *J. Allergy Clin. Immunol.* (2014), doi:10.1016/j.jaci.2014.02.005.
13. W. S. Putnam, J. Li, J. Haggstrom, C. Ng, S. Kadkhodayan-Fischer, M. Cheu, Y. Deniz, H. Lowman, P. Fielder, J. Visich, A. Joshi, N. S. Jumbe, Use of quantitative pharmacology in the development of HAE1, a high-affinity anti-ige monoclonal antibody *AAPS J.* (2008), doi:10.1208/s12248-008-9045-4.

Reviewers' Comments:

Reviewer #1:

Remarks to the Author:

Comments to the revised manuscript and to the rebuttal:

In general, the manuscript has improved significantly by the modifications made. The authors have addressed the points raised in the rebuttal point by point, commented on most of the points and adjusted their manuscript in many ways.

It is also important that the same group or authors has recently published an article in JACI. This very interesting article aims for the design of disruptive IgE inhibitors based on darpins, hence it is clearly related to the aim of the present manuscript. In the revised manuscript the authors do not mention or cite this work although it represents an excellent example of engineering towards receptor displacement. The authors should definitely relate a further revised manuscript to the recently published work. See also below.

Rebuttal points

Point 1: Against the background of the low displacing activity of omalizumab, engineering of improved versions by evolutive techniques is a promising strategy. Here, the authors describe the development of omalizumab variants with increased displacing activity and efficacy. Authors have established a functional yeast display approach allowing selection of mutants for disruptive efficiency. As outcome of the extensive selection process they describe 2 variants, 813 "small increase", and C02 "five-fold" increase in disruptive activity.

Given that the authors claim to apply a disruptive efficiency stain, this increase does not seem impressive, in particular when considering that 813 is similar to parental omalizumab and the well established HAE variant, which is less disruptive than C02 but still more than 813. Another omalizumab variant described previously (i.a. by Davies, 2017, JBC)), also exhibits increased disruptive activity, but is not assessed comparatively in this manuscript and not really discussed at all in this manuscript.

Hence, omalizumab variants with enhanced disruptive activity have been described before, affecting to a certain degree the novelty.

In this work we wanted to modulate the efficiency and potency of disruption to help understand fundamental properties of disruptive antibodies. These studies, to our knowledge, are the first studies that attempt to improve disruption directly, and the first to demonstrate that disruption and affinity can be engineered separately. While a 10-fold improvement in disruptive potency and a 5-fold improvement in efficiency may be less dramatic than changes observed in other parameters in other protein engineering manuscripts (e.g. affinity), we feel that the lack of improvement may be intrinsic to disruption.

In particular, in this specific case of omalizumab, we believe that limitations in disruptive efficiency likely arise because of structural constraints imposed by the omalizumab:IgE interaction. Without greatly changing the antibody binding footprint or binding pose on IgE, the predicted molecular overlap with the receptor remains constant and likely imposes a significant barrier to disruption. These engineering studies modulate how an antibody targets a trimolecular complex while still retaining the antibodies binding to the original epitope and we observe that increasing flexibility within the VH-VL packing plays a significant role in impact disruption. Importantly, even these relatively small changes have the potential to alter the function of these antibodies at relevant doses in human cells (Figure 7).

We would also like note that we were also able to dramatically reduce disruptive efficiency by engineering interfaces unique to the predicted trimolecular complex using the locked IgE:receptor complex (e.g. clone 7 in figure 5i). These observations show conclusively that disruptive efficiency and potency can vary dramatically across antibodies with similar affinities for IgE. These observations also demonstrate that disruptive antibodies have unique non-obvious features that can be engineered to modulate function.

The reviewer is correct to mention the excellent work by Davies et al. They described novel variants and scFv variants of omalizumab with ~2 fold improved affinity (1, 2). However little

quantitative work was published about the ID50 of disruption or disruption efficiency in this work. We estimate that these variants exhibited a ~2-fold improved disruptive potency from single concentration SPR experiments, approximately in proportion to their improved affinity and therefore mirror our results with scFv of omalizumab closely. We have acknowledged this work now early in the results section when discussing scFv studies.

While we did not produce all variants from Davies et al. we did produce the external HAE variant previously engineered for high affinity IgE binding. We would like to highlight that prior to our current manuscript no description of HAE's disruptive activity or potency is present in the literature and it may not have been appreciated by scientists involved in its development. Our HAE observations therefore represent a novel contribution and highlight a previously unknown facet of HAE that we believe is critically important to understanding its adverse effects and failure in a phase 2 clinical trial.

The authors mainly argue with the basic aim of improving the understanding of disruptive antibodies. This understanding is supposed to benefit disruptive antibodies against several kinds of other targets, as briefly mentioned now in the text.

It is reasonable and interesting to include e.g. the HAE antibody for comparative analysis. However, the inclusion of the omalizumab variant from Davies et al. would have been likewise obvious. The authors mainly distract from the point. There is more data on the omalizumab variant available and even PCA data in the patent showing the efficacy of the disruption in vivo. (patent WO2017211928A1). The authors should acknowledge this work.

As the authors state below that the focus on omalizumab makes comparative assessment to darpin derivatives irrelevant in the context of this manuscript, such comparison had been even more important.

Point 2: The abstract claims that the presented work provide a framework for engineering disruptive inhibitors for other targets. It would then be appropriate to provide some examples of other protein based inhibitors in the discussion for which the described procedure may be adapted to. As it stands, the manuscript is too specific to claim general usability as experimental data is only presented for a single system.

We appreciate the reviewers point and removed the claim that the system is "generalizable" in the abstract, while keeping the statement that the system can "provide a conceptual framework for engineering new disruptive inhibitors for other targets." We then expanded in the discussion to explain why we think the system should be generalizable and provide an example of a complex that could be targeted in this system by highlighting a known disruptor of CD47:SIRPa (also discussed in Figure 1d).

The change in wording is acceptable. Adjusting the title of the manuscript might additionally be needed to address the issue appropriately. The title now "Directed evolution of and structural insights into antibody-mediated disruption of stable receptor-ligand complexes" suggest an investigation of multiple receptor ligand complexes, which is not the case. Instead, the singular "of a stable receptor ligand complex" or a specification of which complex is meant, should be used.

Point 3: Related to this, the number of references is unusually low for such a comprehensive and interdisciplinary set of experiments. As an example, three references are given in the materials and methods section. Hence, there is a significant risk that part of the experimental work cannot be reproduced outside the author research team. In addition, not a single reference is listed in the discussion, and the discussion is unusually short. This is very unfortunate, a much more elaborate comparison with prior work is to be expected.

We have added additional citations to the discussion and text where appropriate, expanded the discussion to address several important issues raised by the reviewer (see below), and removed a citation in the methods addressing the production of a protein species (and added in a summary of the cited methods). The other 2 citations in the methods are in reference to kinetic models used to fit binding data and how to program these models into existing software packages. Here the papers provide extensive detailed methods, models, and in one case screen shots of the software interface to facilitate reproduction of the experiments- and we feel would add unnecessary length. We have also added in citations for programs used in methods where relevant.

The authors have adjusted the manuscript by addition of some references in the methods. While the discussion has been extended, only 3 new references have been added, all referring to the concentration of anti-IgE in serum. As pointed out in the initial comment, a more elaborate discussion is important.

As example: a specific topic of relevance in the manuscript is the relation of dwelling time and the risk of receptor cross-linking and anaphylaxis. This risk will for sure be different for other examples mentioned by the authors. What could be the impact of dwelling time in other receptor/ligand complexes?

Moreover, the modified text in the discussion describes the application for de novo disruptive agents. According to the selection procedure, however starting from ligands with known disruptive functions is important for success. This needs to be explained more clearly.

Point 4: The authors relate the variants strongly to the darpins in Figs 2 and 3. This comparison should also be applied in later analysis, e.g. in Fig 7. How would a darpin and a bivalent darpin Fc variant perform in the cellular tests?

We appreciate this point; however we have previously published extensive studies on the function of E2_79 and other bivalent DARPins. While E2_79 and E3_53 were an important control in the protein engineering phase, the aim of this paper was to engineer and enhance omalizumab antibody function. We feel that the data presented clearly demonstrate these points and do not feel that a lack of comparison across all known anti-IgE agents detracts from these observations. As stated above: in the light of the recent paper in JACI a comparative experiment would be even more interesting and important to do.

Point 5: Fig2g-j : darpin and omalizumab seem to behave similarly, which should not be the case

These figures highlight several different assays including: 1. a pure disruption screen, 2. our efficiency screen, and 3. Our co-binding screen. In figure 2 g and h(left panel) the control reagents are exposed to intact complexes alone. This assay most closely reflects disruptive potency and can differentiate E2_79 and omalizumab to an extent (although the magnitude of the difference is small compared to the very large binding signal of E3_53 in the accompanying figures). However this system is attempting to recreate a phenomenon observed with soluble anti-IgE molecules at high concentrations with tethered anti-IgE molecules on the surface of yeast. Therefore it is not surprising that the system displays differences to soluble protein disruption assays. Notably the efficiency stain 2h (right panel) clearly shows how the addition of the G335C-IgE improves the discrimination of these controls. This reagent was primarily used to a control for free IgE affinity during selections to drive disruptive efficiency and in these experiments show how this additional dimension improves the segregation of E2_79 and omalizumab which each have distinct disruptive efficiencies.

Finally in figure 2J we are evaluating how much FcεRIα remains bound to IgE species captured by yeast. In this assay we are trying to distinguish disruptive agents from non-competitive agents and we would expect E2_79 and omalizumab to appear similar as each should primarily disrupt complexes and promote the dissociation of FcεRIα.

Nevertheless we feel that the reviewers' comments here and elsewhere highlight deficiencies in the text and figures in describing selections and we have attempted to clarify these points with additional labels and revised text.

The explanation is helpful, but it still remains inconclusive. For binding of complex in g-right there is virtually no difference between oma and the darpin. Hence, the limited binding in j might reflect no binding instead of disruptive activity.

The data in J appear to be identical to those in 3d. Repetition?

Point 6: Fig 7 : Here a negative control without anti-IgE or with a non-relevant antibody like Humira. Moreover, it would be nice to see Darpin on this assay too.

In figure 7a we have now included an off target anti-CoV2 antibody (D10) and the non-competing DARPIn E3_53 control in the supplemental figure S1 of the paper relating to the introduction to the bead based assay. For figure 7d we have repeated BLI experiments with the corresponding control D10-IgG and DARPIn E3_53 controls and included these in Figure S5I. Finally we have also

included the Humira negative control and crosslinking anti-IgE Le27 positive control for both BMMC and whole human blood anaphylactogenicity experiments in figure 7b and c.

We have published extensively characterizing the DARPin in human cells and BMMCs. While a head-to-head comparison would have been interesting, we feel it is not the primary aim of the paper which focuses on the evolution of omalizumab.

The response is somewhat confusing: in figure 7a there is no change, but in figure S1d there is D10 and a non-disruptive darpin are included. The darpin however shows displacement to a certain degree which should be explained.

Moreover, additional BLI data should be found in S5I, but there is no figure nor data.

The cross-linking Le27 control should be explained in the manuscript, too.

Specific points

P29. "Detailed views of VH mutations (blue) aligned by variant to the native omalizumab scFv (grey)" . IgE is colored grey, what is the meaning?

P33 mid. model building is perhaps in coot as there is no interactive model building program in phenix,

P34 "Carbohydrates were then automatically built into density maps when possible and were not subject to further refinement (Coot)"

Change to "Carbohydrates were then automatically built into EM maps in Coot (ref to Coot) when possible and were not subject to further refinement"

Strictly speaking, the EM map is not a (electron) density, it is the Coulomb potential

Table S5. Cell angles should be with greek letters rather than as a,b,g

The Rwork/Rfree values format need to be unified such that the value without parenthesis and the corresponding value with parenthesis are on the same line in the table as is already the case for the first structure

Reviewer #2:

Remarks to the Author:

The authors have satisfactorily addressed the reviewer's comments.

Reviewer #3:

Remarks to the Author:

I appreciate the additional information provided by the authors. The revised manuscript is even stronger than the original. I was disappointed by one comment in the response to my review, and by the authors' decision not to follow up on it:

Reviewer comment:

- More detail is needed on the properties of the library created by error-prone PCR. In particular, what was the distribution of the location and the number of mutations relative to the parent clone?

Author response:

We did not conduct deep sequencing of the libraries, and given that error prone PCR was employed we can only estimate the frequency of mutations predicted by the protocols associated with the EP polymerase. The mutations should have been randomly distributed throughout the amplified scFv construct.

I suggest that deep sequencing of the libraries be performed and the results included in Supplementary Materials. I disagree that the authors "can only estimate the frequency of mutations" – they can quantify this frequency using next generation sequencing, which is now standard in the field.

Reviewer #4:
None

REVIEWER COMMENTS

We thank the reviewers for their positive comments and assistance in improving the manuscript and hope we have addressed the most recent points as detailed below. For clarity, we have highlighted the new points in bold type, older comments in grey and our previous responses in italics. Our responses to the current points are shown in cyan.

Reviewer #1 (Remarks to the Author):

Comments to the revised manuscript and to the rebuttal:

In general, the manuscript has improved significantly by the modifications made. The authors have addressed the points raised in the rebuttal point by point, commented on most of the points and adjusted their manuscript in many ways.

It is also important that the same group or authors has recently published an article in JACI. This very interesting article aims for the design of disruptive IgE inhibitors based on darpins, hence it is clearly related to the aim of the present manuscript. In the revised manuscript the authors do not mention or cite this work although it represents an excellent example of engineering towards receptor displacement. The authors should definitely relate a further revised manuscript to the recently published work. See also below.

We have added a reference to this work in the introduction (page 4), which was accepted for publication during the submission and revision process of this manuscript and is now only available as a preprint online. Furthermore, we would like to highlight the similar assay formats and controls (e.g. omalizumab) across each paper. We feel that these are very useful for comparing the relative potency and effect of disruptive DARPins. However, we would also like to note that this new manuscript focused on engineering the anchoring domain of a biparatopic molecule. This route to enhanced disruption relies on localizing a potent disruptor at high concentrations adjacent to the IgE:receptor interface via a second non-competitive anti-IgE molecule to promote disruption. In contrast the work here is focused on how a disruptive antibody, without anchoring, could be evolved to enhance disruption.

Rebuttal points

Point 1: Against the background of the low displacing activity of omalizumab, engineering of improved versions by evolutive techniques is a promising strategy. Here, the authors describe the development of omalizumab variants with increased displacing activity and efficacy. Authors have established a functional yeast display approach allowing selection of mutants for disruptive efficiency. As outcome of the extensive selection process they describe 2 variants, 813 “small increase”, and C02 “five-fold” increase in disruptive activity. Given that the authors claim to apply a disruptive efficiency stain, this increase does not seem impressive, in particular when considering that 813 is similar to parental omalizumab and the well established HAE variant, which is less disruptive than C02 but still more than 813. Another omalizumab variant described previously (i.a. by Davies, 2017, JBC)), also exhibits increased disruptive activity, but is not assessed comparatively in this manuscript and not really discussed at all in this manuscript. Hence, omalizumab variants with enhanced disruptive activity have been described before, affecting to a certain degree the novelty.

In this work we wanted to modulate the efficiency and potency of disruption to help understand fundamental properties of disruptive antibodies. These studies, to our knowledge, are the first studies that attempt to improve disruption directly, and the first to demonstrate that disruption and affinity can be engineered separately. While a 10-fold improvement in disruptive potency and a 5-fold improvement in efficiency may be less dramatic than changes observed in other parameters in other protein engineering manuscripts (e.g. affinity), we feel that the lack of improvement may be intrinsic to disruption.

In particular, in this specific case of omalizumab, we believe that limitations in disruptive efficiency likely arise because of structural constraints imposed by the omalizumab:IgE interaction. Without greatly changing the antibody binding footprint or binding pose on IgE, the predicted molecular overlap with the receptor remains constant and likely imposes a significant barrier to disruption. These engineering studies modulate how an antibody targets a trimolecular complex while still retaining the antibodies binding to the original epitope and we observe that increasing flexibility within the VH-VL packing plays a significant role in impact disruption. Importantly, even these relatively small changes have the potential to alter the function of these antibodies at relevant doses in human cells (Figure 7).

We would also like note that we were also able to dramatically reduce disruptive efficiency by engineering interfaces unique to the predicted trimolecular complex using the locked IgE:receptor complex (e.g. clone 7 in figure 5i). These observations show conclusively that disruptive efficiency and potency can vary dramatically across antibodies with similar affinities for IgE. These observations also demonstrate that disruptive antibodies have unique non-obvious features that can be engineered to modulate function.

The reviewer is correct to mention the excellent work by Davies et al. They described novel variants and scFv variants of omalizumab with ~2 fold improved affinity (1, 2). However little quantitative work was published about the ID50 of disruption or disruption efficiency in this work. We estimate that these variants exhibited a ~2-fold improved disruptive potency from single concentration SPR experiments, approximately in proportion to their improved affinity and therefore mirror our results with scFv of omalizumab closely. We have acknowledged this work now early in the results section when discussing scFv studies.

While we did not produce all variants from Davies et al. we did produce the external HAE variant previously engineered for high affinity IgE binding. We would like to highlight that prior to our current manuscript no description of HAE's disruptive activity or potency is present in the literature and it may not have been appreciated by scientists involved in its development. Our HAE observations therefore represent a novel contribution and highlight a previously unknown facet of HAE that we believe is critically important to understanding its adverse effects and failure in a phase 2 clinical trial.

The authors mainly argue with the basic aim of improving the understanding of disruptive antibodies. This understanding is supposed to benefit

disruptive antibodies against several kinds of other targets, as briefly mentioned now in the text. It is reasonable and interesting to include e.g. the HAE antibody for comparative analysis. However, the inclusion of the omalizumab variant from Davies et al. would have been likewise obvious. The authors mainly distract from the point. There is more data on the omalizumab variant available and even PCA data in the patent showing the efficacy of the disruption in vivo. (patent WO2017211928A1). The authors should acknowledge this work. As the authors state below that the focus on omalizumab makes comparative assessment to darpin derivatives irrelevant in the context of this manuscript, such comparison had been even more important.

We feel it is very important to distinguish peer reviewed publications and patents, and do not think citing patents in peer reviewed documents is essential or appropriate. In the peer reviewed literature the quantitative analysis of the variants produced by Davies et al. across multiple publications is limited. However we have cited this work to allow the reader to compare the bodies of work and appreciate the overlapping and distinct observations on omalizumab MOA.

More importantly, our manuscript does not claim to have generated the best omalizumab variant ever- nor is it the point of the work. Multiple efforts have been undertaken to improve the potency and affinity of omalizumab (e.g. Xencor AIMab7195, Genetech HAE, and Davies et. al.). Rather HAE was included in the manuscript precisely because it was affinity matured and therefore served as a benchmark for affinity maturation as compared to the efficiency selections presented here. Re-benchmarking multiple antibody variants across multiple assays without any clear new mechanistic or scientific purpose would not add to the manuscript. HAE was also interesting given its history in clinical trials that demonstrated safety concerns – which we have now correlated with longer dwell time on complexes. We noted that some of our variants that were slow to disrupt induced spontaneous activation and highlighted what may be an important parameter of disruptive anti-IgE antibodies. Furthermore, this work describes for the first time the previously unknown disruptive potency of HAE.

In regard to DARPins, as previously mentioned, there are many similarly executed basophil, mast cell, or bead-based disruption assays in our recent JACI publication and also prior JACI publications that allow for comparison between the inhibitors. As noted, we have added text to include a comparative assessment of these reagents (page 4), and to help illustrate the important consequences of adding anchoring domains to disruptive inhibitors. However, repeating published experiments to rank anti-IgE molecules is not the aim of this paper.

Point 2: The abstract claims that the presented work provide a framework for engineering disruptive inhibitors for other targets. It would then be appropriate to provide some examples of other protein based inhibitors in the discussion for which the described procedure may be adapted to. As it stands, the manuscript is too specific to claim general usability as experimental data is only presented for a single system.

We appreciate the reviewers point and removed the claim that the system is “generalizable” in the abstract, while keeping the statement that the system can “provide a conceptual framework for engineering new disruptive inhibitors for other targets.” We then expanded in the discussion to explain why we think the system should be generalizable and provide an example of a complex that could be targeted in this system by highlighting a known disruptor of CD47:SIRPa (also discussed in Figure 1d).

The change in wording is acceptable. Adjusting the title of the manuscript might additionally be needed to address the issue appropriately. The title now “Directed evolution of and structural insights into antibody-mediated disruption of stable receptor-ligand complexes” suggest an investigation of multiple receptor ligand complexes, which is not the case. Instead, the singular “of a stable receptor ligand complex” or a specification of which complex is meant, should be used.

We have changed the title to “**Directed evolution of and structural insights into antibody-mediated disruption of a stable receptor-ligand complex.**”

Point 3: Related to this, the number of references is unusually low for such a comprehensive and interdisciplinary set of experiments. As an example, three references are given in the materials and methods section. Hence, there is a significant risk that part of the experimental work cannot be reproduced outside the author research team. In addition, not a single reference is listed in the discussion, and the discussion is unusually short. This is very unfortunate, a much more elaborate comparison with prior work is to be expected.

We have added additional citations to the discussion and text where appropriate, expanded the discussion to address several important issues raised by the reviewer (see below), and removed a citation in the methods addressing the production of a protein species (and added in a summary of the cited methods). The other 2 citations in the methods are in reference to kinetic models used to fit binding data and how to program these models into existing software packages. Here the papers provide extensive detailed methods, models, and in once case screen shots of the software interface to facilitate reproduction of the experiments- and we feel would add unnecessary length. We have also added in citations for programs used in methods where relevant.

The authors have adjusted the manuscript by addition of some references in the methods. While the discussion has been extended, only 3 new references have been added, all referring to the concentration of anti-IgE in serum. As pointed out in the initial comment, a more elaborate discussion is important.

As example: a specific topic of relevance in the manuscript is the relation of dwelling time and the risk of receptor cross-linking and anaphylaxis. This risk will for sure be different for other examples mentioned by the authors. What could be the impact of dwelling time in other receptor/ligand complexes?

Moreover, the modified text in the discussion describes the application for de novo disruptive agents. According to the selection procedure, however starting from ligands with known disruptive functions is important for success. This needs to be explained more clearly.

The reviewer brings up an important point that efficiency and dwell time will be important in some biologic systems and irrelevant in others. This is a very broad topic and would be largely speculative but we have modified the text relevant to this as follows:

Original: "These reagents have been described for many receptor ligand pairs, such as CD47 and SIRP α (30), and antibodies that can disrupt CD47:SIRP α complexes have already been described (5). Therefore we anticipate that multiple protein systems should be amenable to the isolation and engineering of disruptive inhibitors."

New: "These reagents have been described for many receptor ligand pairs, such as CD47 and SIRP α (30), and antibodies that can disrupt CD47:SIRP α complexes have already been described (5). We anticipate that multiple protein systems should be amenable to the isolation and engineering of disruptive inhibitors, although the importance of disruptive efficiency and dwell time of disruptive molecules will depend on the underlying biology of the system."

While we did not selected for a disruptive agent de novo, we have shown that we can classify good disruptors, weak disruptors, and non-competitive agents with a range of affinities for IgE. Therefore, the assay can clearly discriminate between these classes of antibodies. We do extract better disruptors out of a library of mutated omalizumab variants, demonstrating the ability to provide a finer discrimination between these activities than would be required for the isolation of novel inhibitors, although technically that has yet to be demonstrated. To address this directly in our current manuscript, we have clarified the discussion point with the following sentence: "In this manuscript we have not directly isolated novel disruptors from a library of binders, yet we can classify a range of anti-IgE agents with distinct epitopes, affinities, and disruptive potencies suggesting the techniques could be used for de novo selections."

Point 4: The authors relate the variants strongly to the darpins in Figs 2 and 3. This comparison should also be applied in later analysis, e.g. in Fig 7. How would a darpin and a bivalent darpin Fc variant perform in the cellular tests?

We appreciate this point; however we have previously published extensive studies on the function of E2_79 and other bivalent DARPins. While E2_79 and E3_53 were an important control in the protein engineering phase, the aim of this paper was to engineer and enhance omalizumab antibody function. We feel that the data presented clearly demonstrate these points and do not feel that a lack of comparison across all known anti-IgE agents detracts from these observations.

As stated above: in the light of the recent paper in JACI a comparative experiment would be even more interesting and important to do.

Our publication in JACI is focused on understanding how the anchoring domain of a bi-paratopic disruptive inhibitor can be modified to improve overall activity. This is a fundamentally different focus than understanding how a disruptive inhibitor or inhibitory domain can be optimized on its own, which is the subject of this work. Here we have demonstrated how one can evolve the omalizumab Fab to increase its disruptive activity to a level that is comparable to the inhibitory arm of the bi-paratopic DARPin in the JACI paper. Anchoring our Fabs to the IgE:FceRI complex by making biparatopic antibodies or antibody:DARPin hybrids will improve the activity and make for an interesting comparison with the JACI molecules, but that is beyond the scope of this work. As mentioned above we have added text to cite the comparison. In addition, the comparison of these reagents should be very straightforward using the published data, given the many similar functional assays across the two bodies of work. Repeating them here is beyond the scope of the paper, and we are confident that our reported IC50s in biochemical and cell-based assays provide an excellent basis for comparison.

Point 5: Fig2g-j : darpin and omalizumab seem to behave similarly, which should not be the case

These figures highlight several different assays including: 1. a pure disruption screen, 2. our efficiency screen, and 3. Our co-binding screen. In figure 2g and h(left panel) the control reagents are exposed to intact complexes alone. This assay most closely reflects disruptive potency and can differentiate E2_79 and omalizumab to an extent (although the magnitude of the difference is small compared to the very large binding signal of E3_53 in the accompanying figures). However this system is attempting to recreate a phenomenon observed with soluble anti-IgE molecules at high concentrations with tethered anti-IgE molecules on the surface of yeast. Therefore it is not surprising that the system displays differences to soluble protein disruption assays. Notably the efficiency stain 2h (right panel) clearly shows how the addition of the G335C-IgE improves the discrimination of these controls. This reagent was primarily used to a control for free IgE affinity during selections to drive disruptive efficiency and in these experiments show how this additional dimension improves the segregation of E2_79 and omalizumab which each have distinct disruptive efficiencies.

Finally in figure 2j we are evaluating how much FceRIa remains bound to IgE species captured by yeast. In this assay we are trying to distinguish disruptive agents from non-competitive agents and we would expect E2_79 and omalizumab to appear similar as each should primarily disrupt complexes and promote the dissociation of FceRIa.

Nevertheless we feel that the reviewers' comments here and elsewhere highlight deficiencies in the text and figures in describing selections and we have attempted to clarify these points with additional labels and revised text.

The explanation is helpful, but it still remains inconclusive. For binding of complex in g-right there is virtually no difference between oma and the darpin. Hence, the limited binding in j might reflect no binding instead of disruptive activity. The data in J appear to be identical to those in 3d. Repetition?

We would like to thank the reviewer for catching our error in figure 2j and 3d. We have revised the figures to include a separate additional experiment and did not intend to repeat these histograms. Furthermore, to address the reviewer's concern about the possibility of no IgE binding we have included an additional blank yeast control exposed to all secondary reagents but not IgE or IgE:FcR complexes to demonstrate that each agent is indeed binding IgE. Below we have also analyzed the uninduced, cMyc-negative yeast within the same tube as the displayed anti-IgE agent,

which are exposed to all the same reagents. These yeast provide an internal background control for all of the staining reagents, as they only differ in the lack of induction of the anti-IgE protein of interest. These data further show that the IgE binding is specific to the displayed control proteins. We have noted the specificity of the staining in our manuscript figure legend but left out the cMyc- controls in the new figure panel given space constraints.

Finally given the confusion surrounding figure 2g we have opted to add back the axis to the R panel and adjust the range to better visualize the separation. Yeast induction is variable across days and constructs- and the separation of these two controls is perhaps less than would be expected from other biochemical assays. But the separation is clear and becomes much more evident in dual color assay panels. These minor differences can also be appreciated in our new figure 2J panel. As mentioned before, translating biochemical phenomenon to yeast display will not be a perfect 1:1 translation.

Point 6: Fig 7 : Here a negative control without anti-IgE or with a non-relevant antibody like Humira. Moreover, it would be nice to see Darpin on this assay too.

In figure 7a we have now included an off target anti-CoV2 antibody (D10) and the non-competing DARPin E3_53 control in the supplemental figure S1 of the paper relating to the introduction to the bead based assay. For figure 7d we have repeated BLI experiments with the corresponding control D10-IgG and DARPin E3_53 controls and included these in Figure S51. Finally we have also included the Humira negative control and crosslinking anti-IgE Le27 positive control for both BMMC and whole human blood anaphylactogenicity experiments in figure 7b and c.

We have published extensively characterizing the DARPins in human cells and BMMCs. While a head-to-head comparison would have been interesting, we feel it is not the primary aim of the paper which focuses on the evolution of omalizumab.

The response is somewhat confusing: in figure 7a there is no change, but in figure S1d there is D10 and a non-disruptive darpin are included. The darpin however shows displacement to a certain degree which should be explained.

Moreover, additional BLI data should be found in S51, but there is no figure nor data.

The cross-linking Le27 control should be explained in the manuscript, too.

For clarity we reorganized the supplemental figures and S51 is now S7g- but our previous reviewer response text was not corrected to reflect this change in the figures. We have also added text to cite our new JACI publication and explain the partial displacement by E3_53 in figure S1d on page 5 (this result is discussed in detail in the JACI publication and is likely secondary to allosteric modulation of the IgE-Fc through interactions at the Ce2-Ce3 interface). Finally we added text to clarify the cross-linking anti-IgE Le27 control (in figure legend 7 and methods).

Specific points

P29. "Detailed views of VH mutations (blue) aligned by variant to the native omalizumab scFv (grey)" . IgE is colored grey, what is the meaning?

This coloring scheme was used to be consistent across figures, and to pick a color that stood out from the yellow and blue VH and VL domains. There is no meaning. We adjusted figure legend to note dark grey and light grey for each.

P33 mid. model building is perhaps in coot as there is no interactive model building program in phenix,

We have noted this and added a citation for coot distributed with phenix to give proper credit to the coot developers.

**P34 “Carbohydrates were then automatically built into density maps when possible and were not subject to further refinement (Coot)”
Change to “Carbohydrates were then automatically built into EM maps in Coot (ref to Coot) when possible and were not subject to further refinement”**

Changed as requested.

Strictly speaking, the EM map is not a (electron) density, it is the Coulomb potential

We could not find an instance of “electron density,” in our manuscript. Experimental density, map density, etc. are routinely used to describe potential maps in Cryo-EM manuscripts. We are not aware of a better convention and have made sure that we did not describe the density as electron density.

Table S5. Cell angles should be with greek letters rather than as a,b,g

Error fixed as requested.

The Rwork/Rfree values format need to be unified such that the value without parenthesis and the corresponding value with parenthesis are on the same line in the table as is already the case for the first structure

Changed as requested.

Reviewer #2 (Remarks to the Author):

The authors have satisfactorily addressed the reviewer's comments.

Reviewer #3 (Remarks to the Author):

I appreciate the additional information provided by the authors. The revised manuscript is even stronger than the original. I was disappointed by one comment in the response to my review, and by the authors’ decision not to follow up on it:

Reviewer comment:

- More detail is needed on the properties of the library created by error-prone PCR. In particular, what was the distribution of the location and the number of mutations relative to the parent clone?

Author response:

We did not conduct deep sequencing of the libraries, and given that error prone PCR was employed we can only estimate the frequency of mutations predicted by the protocols associated with the EP polymerase. The mutations should have been randomly distributed throughout the amplified scFv construct.

I suggest that deep sequencing of the libraries be performed and the results included in Supplementary Materials. I disagree that the authors “can only estimate the frequency of mutations” – they can quantify this frequency using next generation sequencing, which is now standard in the field.

We did not mean to imply that one couldn’t estimate the frequency of mutations - we just meant that we did not estimate this frequency. We do not see the scientific insight that would be gained by deep sequencing these error prone libraries as we did not use sequence data to inform selections until we began screening clones. If the diversity was much smaller than estimated by transformants it would simply mean one could make larger libraries and find more mutations. If the diversity was larger perhaps our selections would be more exhaustive than we estimated. However, given the enormous sequence space any quantitative answer we supply would still suggest more mutations could be made. Furthermore a literature search quickly points out that many papers employing yeast display do not deep sequence libraries when the deep sequencing is not being used to drive decision making (e.g. <https://www.nature.com/articles/s41467-021-21609-2#Sec17> standard sequencing of clones).

Reviewers' Comments:

Reviewer #1:

Remarks to the Author:

The authors have addressed most of the points appropriately and adjusted the manuscript according to the suggestions. As outlined below minor corrections remain to be done.

Author: We have added a reference to this work in the introduction (page 4), which was accepted for publication during the submission and revision process of this manuscript and is now only available as a preprint online. Furthermore, we would like to highlight the similar assay formats and controls (e.g. omalizumab) across each paper. We feel that these are very useful for comparing the relative potency and effect of disruptive DARPins. However, we would also like to note that this new manuscript focused on engineering the anchoring domain of a biparatopic molecule. This route to enhanced disruption relies on localizing a potent disruptor at high concentrations adjacent to the IgE:receptor interface via a second non-competitive anti-IgE molecule to promote disruption. In contrast the work here is focused on how a disruptive antibody, without anchoring, could be evolved to enhance disruption.

The mentioned reference has been added as under revision. As the authors state, the manuscript has been accepted and is publicly available. Please adjust accordingly.

Author: We feel it is very important to distinguish peer reviewed publications and patents, and do not think citing patents in peer reviewed documents is essential or appropriate. In the peer reviewed literature the quantitative analysis of the variants produced by Davies et al. across multiple publications is limited. However we have cited this work to allow the reader to compare the bodies of work and appreciate the overlapping and distinct observations on omalizumab MOA.

It is correct that patents are not peer reviewed publications and therefore citing patents might not be essential. Whenever patents can provide knowledge that helps the reader to understand the field, citing is for sure appropriate. Hence I still recommend to add the citation.

The reference to the published data by Davies et al (15) is identical to reference 28. Please adjust accordingly.

Author : In regard to DARPins, as previously mentioned, there are many similarly executed basophil, mast cell, or bead-based disruption assays in our recent JACI publication and also prior JACI publications that allow for comparison between the inhibitors. As noted, we have added text to include a comparative assessment of these reagents (page 4), and to help illustrate the important consequences of adding anchoring domains to disruptive inhibitors. However, repeating published experiments to rank anti-IgE molecules is not the aim of this paper.

The authors are obviously correct that ranking is not the aim of the paper. It is however a matter of congruency if disruption efficacy initially is compared to the darpin and other molecules and later on in the manuscript a comparison is not pursued but considered a ranking.

Author: Our publication in JACI is focused on understanding how the anchoring domain of a bi-paratopic disruptive inhibitor can be modified to improve overall activity. This is a fundamentally different focus than understanding how a disruptive inhibitor or inhibitory domain can be optimized on its own, which is the subject of this work. Here we have demonstrated how one can evolve the omalizumab Fab to increase its disruptive activity to a level that is comparable to the inhibitory arm of the bi-paratopic DARPIn in the JACI paper. Anchoring our Fabs to the IgE:FcεRI complex by making biparatopic antibodies or antibody:DARPIn hybrids will improve the activity and make for an interesting comparison with the JACI molecules, but that is beyond the scope of this work. As mentioned above we have added text to cite the comparison. In addition, the comparison of these reagents should be very straightforward using the published data, given the many similar

functional assays across the two bodies of work. Repeating them here is beyond the scope of the paper, and we are confident that our reported IC50s in biochemical and cell-based assays provide an excellent basis for comparison.

As above.

Point By Point Response Below:

REVIEWERS' COMMENTS

Reviewer #1 (Remarks to the Author):

The authors have addressed most of the points appropriately and adjusted the manuscript according to the suggestions. As outlined below minor corrections remain to be done.

Author: We have added a reference to this work in the introduction (page 4), which was accepted for publication during the submission and revision process of this manuscript and is now only available as a preprint online. Furthermore, we would like to highlight the similar assay formats and controls (e.g. omalizumab) across each paper. We feel that these are very useful for comparing the relative potency and effect of disruptive DARPins. However, we would also like to note that this new manuscript focused on engineering the anchoring domain of a biparatopic molecule. This route to enhanced disruption relies on localizing a potent disruptor at high concentrations adjacent to the IgE:receptor interface via a second non-competitive anti-IgE molecule to promote disruption. In contrast the work here is focused on how a disruptive antibody, without anchoring, could be evolved to enhance disruption.

The mentioned reference has been added as under revision. As the authors state, the manuscript has been accepted and is publicly available. Please adjust accordingly.

PBP Response: We have now updated this.

Author: We feel it is very important to distinguish peer reviewed publications and patents, and do not think citing patents in peer reviewed documents is essential or appropriate. In the peer reviewed literature the quantitative analysis of the variants produced by Davies et al. across multiple publications is limited. However we have cited this work to allow the reader to compare the bodies of work and appreciate the overlapping and distinct observations on omalizumab MOA.

It is correct that patents are not peer reviewed publications and therefore citing patents might not be essential. Whenever patents can provide knowledge that helps the reader to understand the field, citing is for sure appropriate. Hence I still recommend to add the citation.

The reference to the published data by Davies et al (15) is identical to reference 28. Please adjust accordingly.

PBP Response: We have revised the citation for Davie et al and feel that it is more appropriate for a journal publication to leave the non-peer reviewed work uncited. Patents have not undergone the same level of scientific scrutiny as the scientific literature and we do not feel that it is correct to include this citation. Patents contain claims that may be informative but that may also be misleading if they have not been given sufficient scrutiny by peer reviewers the ensure that the scientific foundation for the claims is justified.

Author : In regard to DARPins, as previously mentioned, there are many similarly executed basophil, mast cell, or bead-based disruption assays in our recent JACI publication and also prior JACI publications that allow for comparison between the inhibitors. As noted, we have added text to include a comparative assessment of these reagents (page 4), and to help illustrate the important consequences of adding anchoring domains to disruptive inhibitors. However, repeating published experiments to rank anti-IgE molecules is not the aim of this paper.

The authors are obviously correct that ranking is not the aim of the paper. It is however a matter of congruency if disruption efficacy initially is compared to the darpin and other molecules and later on in the manuscript a comparison is not pursued but considered a ranking.

PBP Response: Per the journals and reviewers request we have supplied a table with relative potency of each molecule in assays that were replicated and similar across manuscripts. In this way we can provide a quantitative comparison across separate experimental replicates in a matched assay.

In our human and BMCM studies we do have sufficient data points to accurately assign an in vitro KD on similar time scales- however the bead assay data above fairly ranks each molecule by potency and emphasizes the very large improvements in potency observed in biparatopic anchored DARPins anchored anti-IgE molecules.

This point is highlighted in a new sentence in the discussion and new table #S11

Author: Our publication in JACI is focused on understanding how the anchoring domain of a bi-paratopic disruptive inhibitor can be modified to improve overall activity. This is a fundamentally different focus than understanding how a disruptive inhibitor or inhibitory domain can be optimized on its own, which is the subject of this work. Here we have demonstrated how one can evolve the omalizumab Fab to increase its disruptive activity to a level that is comparable to the inhibitory arm of the bi-paratopic DARPin in the JACI paper. Anchoring our Fabs to the IgE:FcεRI complex by making biparatopic antibodies or antibody:DARPin hybrids will improve the activity and make for an interesting comparison with the JACI molecules, but that is beyond the scope of this work. As mentioned above we have added text to cite the comparison. In addition, the comparison of these reagents should be very straightforward using the published data, given the many similar functional assays across the two bodies of work. Repeating them here is beyond the scope of the paper, and we are confident that our reported IC50s in biochemical and cell-based assays provide an excellent basis for comparison.

As above.

PBP Response: See above